# A genomic appraisal of invasive *Salmonella* Typhimurium and associated antibiotic resistance in sub-Saharan Africa

Invasive non-typhoidal *Salmonella* (iNTS) disease manifesting as bloodstream infection with high mortality is responsible for a huge public health burden in sub-Saharan Africa. *Salmonella enterica* serovar Typhimurium (*S*. Typhimurium) is the main cause of iNTS disease in Africa. By analysing whole genome sequence data from 1303 *S*. Typhimurium isolates originating from 19 African countries and isolated between 1979 and 2017, here we show a thorough scaled appraisal of the population structure of iNTS disease caused by *S*. Typhimurium across many of Africa's most impacted countries. At least six invasive *S*. Typhimurium clades have already emerged, with ST313 lineage 2 or ST313-L2 driving the current pandemic. ST313-L2 likely emerged in the Democratic Republic of Congo around 1980 and further spread in the mid 1990s. We observed plasmid-borne as well as chromosomally encoded fluoroquinolone resistance underlying emergences of extensive-drug and pan-drug resistance. Our work provides an overview of the evolution of invasive *S*. Typhimurium disease, and can be exploited to target control measures.

Invasive non-typhoidal *Salmonella* (iNTS) disease has emerged as a major cause of morbidity and mortality in sub-Saharan Africa (sSA)[1,2]. There are an estimated 594,000 cases annually with a mean case fatality rate of 14.5%, while these incidence figures may be an underestimation as surveillance for iNTS disease is infrequent[3,4]. This systemic disease predominantly occurs among infants and young children, children with malaria and malnutrition and individuals living with human immunodeficiency virus (HIV) infections[2]. Data on iNTS disease in sSA are still relatively scarce due to limited surveillance for bloodstream infections across the continent but estimated incidences show high variability in regions and countries in sSA, as well as within countries[5,6].

*Salmonella enterica* serovar Typhimurium (*S*. Typhimurium) is the most common cause of iNTS disease in sSA[7]. While *S*. Typhimurium strains causing gastroenteritis globally predominantly fall into multilocus sequence types (MLST) ST19 and ST34[8], strains causing bloodstream infections in sSA are highly associated with clade ST313. These African-associated ST313 broadly fall into distinct lineages known as 1 and 2, and both are associated with multidrug resistance (MDR)[9]. Epidemiological and genomic analysis has shown that lineage 1

emerged first and is predominantly confined to the East African region. Lineage 2 emerged later and has spread broadly across sSA, potentially driven by the HIV epidemic and the use of chloramphenicol[9,10]. However, a strict division between invasive and non-invasive *S*. Typhimurium based on the MLSTs does not appear to represent the situation well. Invasive *S*. Typhimurium disease is not exclusively associated with ST313 in sSA, as in Kenya, ST19 *S*. Typhimurium have also been reported as a relatively common cause of such bloodstream infections[11] and non-lineage 1 and 2 ST313 *S*. Typhimurium isolates isolated in the UK are generally associated with gastro-intestinal disease and present no travel association to sSA[12].

ST313 isolates are typically MDR with some having acquired extended-spectrum beta-lactamase (ESBL) genes driving ceftriaxone resistance, whilst resistance to fluoroquinolones (FQ, both plasmid and chromosomally encoded) and more recently to azithromycin (AZI) is emerging[9,10,13,14]. This development significantly limits treatment options for iNTS disease in low- and middle-income countries (LMICs)[15].

Despite the significant burden of invasive *S*. Typhimurium disease in sSA, our understanding of the bacterial population structure and

✉ e-mail: sandra.vanpuyvelde@uantwerpen.be

consequently epidemiology is poor, which contributes to the lack of appropriate disease interventions. Herein we report on a broad genomic analysis approach involving 1303 *S.* Typhimurium strains, isolated between 1979 and 2017 from 19 sSA countries, providing an optimal update following on from the initial pan-African study by Okoro et al.[9]. Our study yields a timely perspective on the distribution and spread of the invasive *S.* Typhimurium pandemic of sSA including an analysis of antimicrobial resistance, including hotspots defined as geographical foci where extensive drug resistant (XDR) isolates are emerging. These data will yield a better understanding of the disease and will support the targeting of interventions.

## Results

### Several independent clades of invasive *S.* Typhimurium have emerged in Africa

Although whole genome sequencing has been used to analyse iNTS isolates, previous work focused on regional data or on the ST313 clade. To obtain a pan-African perspective, we analysed whole genome sequencing data of 1302 African *S.* Typhimurium in the context of 117 non-African *S.* Typhimurium isolates, including eight African travel-associated isolates. The African isolates span 1979–2017 (Supplementary Fig. 1), with the greatest density of sampling between 2008 and 2017, when 73.8% (961 out of 1302) strains were isolated.

Our phylogenetic analysis captured more population diversity than previously reported[9,16], whereas two 313 lineages 1 and 2 were reported before to cause invasive *S.* Typhimurium infections, we identified 4 additional major ST19 lineages. The majority of isolates (n = 1005) form a tight cluster within ST313 lineage II (ST313-L2), including isolates from East, Central and West African countries (Fig. 1a and Supplementary Fig. 1). ST313 lineage I (ST313-L1) isolates were much less prevalent (n = 87) and originated predominantly from East African countries. An additional 258 African *S.* Typhimurium isolates belonged to ST19 and included four monophyletic clusters (Supplementary Fig. 3).

We identified six major clades associated with invasive *S.* Typhimurium infections in sSA. Four invasive *S.* Typhimurium clades were ST19 (ST19-L1 to L4) while two clades were ST313-L1 and ST313-L2, respectively (Supplementary Fig. 4 and Fig. 1a). Most of the invasive *S.* Typhimurium clades were identified in multiple countries, suggesting a widespread distribution of these clades (Fig. 1b). The East African region showed the highest diversity of clades, with all six invasive *S.* Typhimurium clades being identified. At the country level, Kenya and DRC showed the highest diversity, each with 5 out of 6 of the invasive *S.* Typhimurium clades being identified. Notably, ST19-L2 and ST19-L4 were apparently older clades and were identified in the 1980s in Rwanda and DRC, while ST19-L4 isolates were still being observed among the recent samples and in other regions. ST19-L1 and ST19-L3 were predominantly observed in the 2010s (Fig. 1c). ST313-L1 was observed throughout the 1990s and until recently, while the ST313-L2 isolates bacame most dominant since 2001 and appear to be driving the current pandemic in sSA. The ST313-L2 clade was further classified in seven subclades, (Supplementary Figs. 5 and 6 and Fig. 2a) with each subclade associated to one country, i.e., DRC (n = 5), Kenya (n = 1) and Malawi (n = 1), which suggests local clonal expansions (Fig. 2b).

### Continental spread of the current ST313-L2 clade occurred in 1994–1996

A representative selection of 252 ST313-L2 isolates (Supplementary Fig. 7) was subjected to a phylogeographical analysis in which the country of the ancestral strains was predicted to gather further information on the spread of invasive *S.* Typhimurium across sSA. Our analysis assigned the DRC as the most likely source of ST313-L2, with the most recent common ancestor (MRCA) predicted to have originated in 1980 (95% highest posterior density (HPD) confidence interval 1974–1986), refining the earlier prediction of the ST313-L2 MRCA from

1977 (95% HPD 1957–1988)[9]. We found no evidence that ST313-L2 was present anywhere outside the DRC until 1994, supported by probability rates >97% for ancestry nodes in that period.

From 1994 onwards, we observed five independent introductions from the DRC into the East African region (two times into Uganda in 1996, once to Malawi in 1996, once to Malawi in 2000 and once to Kenya in 1995, all observed in Fig. 3 when the most likely location of a branch changes from DRC to East Africa) and one major introduction in the West African region (to Ghana in 1994) which preceded its subsequent spread. We also identified further introductions from West to East Africa. In West Africa, we detected frequent transmission events across borders, including transmission to Malawi, whereas in East Africa, evidence for spread to neighbouring countries was less apparent (Fig. 3 and Supplementary Fig. 8).

The ten ST313-L2 UK isolates could be directly linked to eight separate transmission events from Africa[12], underlining the importance of travel-associated transmissions of invasive *S.* Typhimurium.

### Multiple chromosomal SNPs conferring resistance to ciprofloxacin have occurred in invasive *S.* Typhimurium

Acquisitions of chromosomal SNPs in quinolone resistance-determining regions (QRDR) were observed across the population and in the different regions of sSA (Supplementary Fig. 9). A total of 17 independent QRDR SNP acquisition events were observed in the African invasive *S.* Typhimurium clades, resulting in substitutions of GyrA S83F (n = 3), GyrA S83Y (n = 4), GyrA D87G (n = 1), GyrA D87N (n = 3), GyrA D87Y (n = 5), GyrB E466Y (n = 1) (Table 1). Eight of these QRDR SNP acquisition events gave rise to clusters of multiple isolates, suggesting local outbreaks. Three of these were observed in Kenya, three in DRC and two in Ghana. One of the Kenyan clusters contained isolates from 2005 to 2014, while one DRC cluster contained 49 isolates isolated between 2013 and 2017 without interruption, both suggesting a long-term circulation of the decreased ciprofloxacin susceptibility (DCS) clusters possibly linked to an ongoing outbreak.

The earliest observation of DCS was in 2002 (DRC, Lwiro), with subsequent acquisitions being continuously observed since then in East (Kenya), Central (DRC) and West (Benin, Ghana) Africa (Fig. 4). QRDR SNP acquisitions originated from different regions across the country and were not notably linked to a single local hotspot in Kenya or DRC (Table 1).

The large GyrA S83Y cluster in ST313-L2 subclade 7 (n = 49) clade showed nested phylogenetic substructures, with significantly increased MIC values for ciprofloxacin in the emerging substructures (P value = 0.0235), starting at a median MIC of 0.125 mg/L and mean MIC of 0.132 mg/L and reaching a median MIC of 0.190 mg/L and mean MIC of 0.188 mg/L (Supplementary Fig. 2). No known genetic mechanisms for ciprofloxacin resistance were identified that explain these subtle increases in ciprofloxacin MIC values. However, non-synonymous SNPs were acquired in virulence gene *speC*, nitroreductase gene *nfsA*, *SBOV13191*, *ydhZ*, ethanolamine reactivase gene *eutA* and malate synthase gene *aceB*, none of which have been implicated in ciprofloxacin resistance before.

### XDR and PDR invasive *S.* Typhimurium is associated with the presence of IncHI2 and IncI1 plasmids

Isolates that carried genetic markers underlying XDR or PDR, including ESBL activity, FQ and AZI resistance, were scattered across the phylogeny (Supplemental Fig. 10). XDR and PDR were observed in four different combinations (Fig. 5). Our data revealed that XDR has emerged at least six times in invasive *S.* Typhimurium, with one emergence underlying ST313-L2 subclade 3[10]. XDR was associated with either acquisition of AZI resistance (*mphA*) and ESBL (*SHV*) markers, or by acquired FQ resistance (*qnr*) and ESBL (*CTX-M*) markers. PDR, on the other hand, was observed three independent times, each time as a combination of AZI resistance (*mphA*), ESBL (*SHV*) and FQ resistance

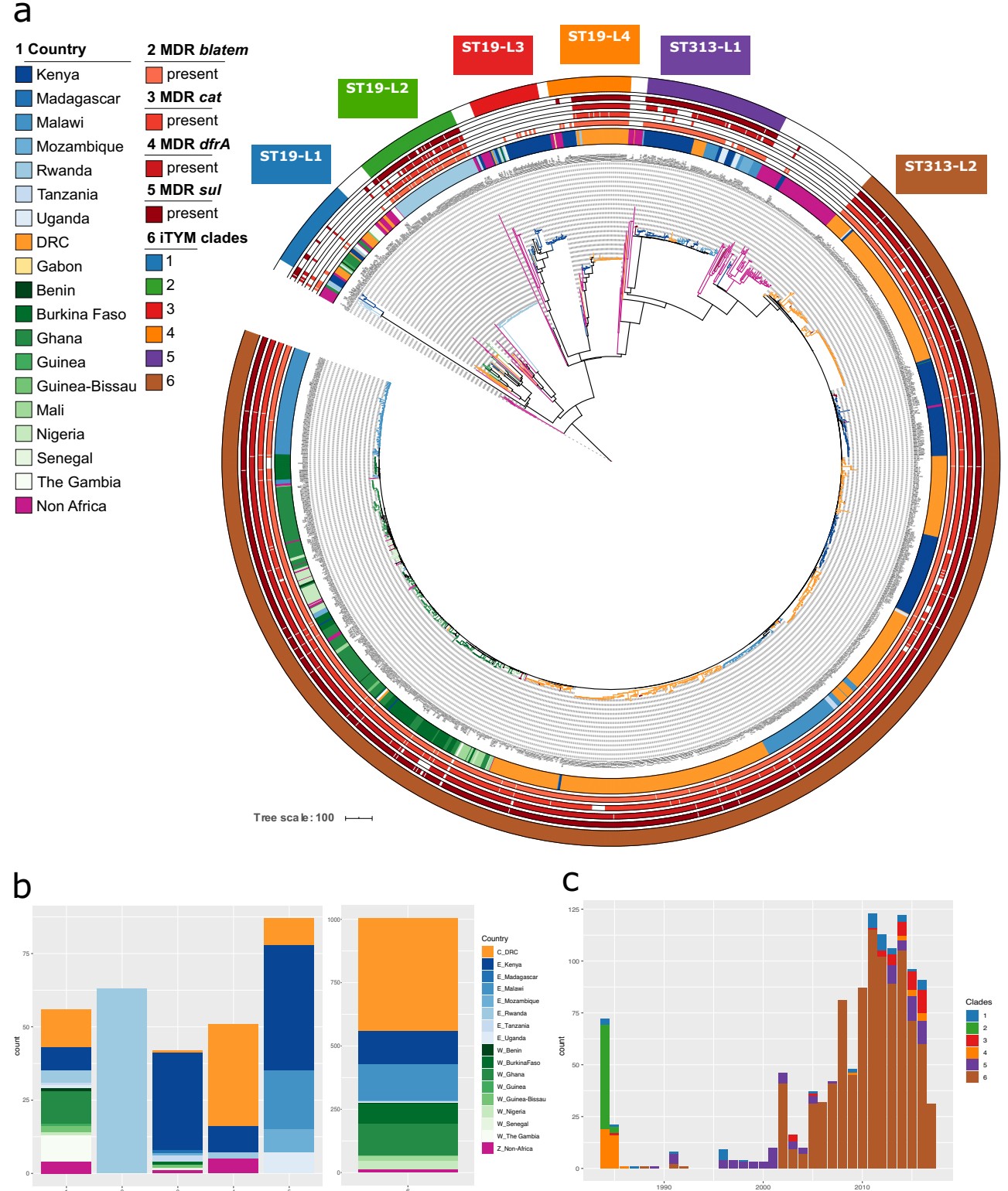

**Fig. 1 | The distribution of invasive *S*. Typhimurium in Africa. a** Maximum likelihood phylogenetic tree of the 1419 *S*. Typhimurium isolates sequences from this study (summarised in Supplementary Data 1). Sequencing reads were mapped to *S*. Typhimurium ST313 reference strain D23580. The tree is based on 71521 chromosomal SNPs. Branches are coloured by the country of isolation. Invasive *S*. Typhimurium clades as identified in this study are annotated. Metadata is visualised on the concentric rings in compliance to the legend, from the inside to outside; (1) Country of origin, (2–5) presence of multidrug resistance markers (MDR; *blaTEM*, *cat*, *dfrA*, *sul*), (6) invasive *S*. Typhimurium clades. Branch lengths represent the number of SNPs as indicated in the scale bar. **b** Distribution of invasive *S*. Typhimurium clades per country for the studied isolates assigned to an invasive *S*. Typhimurium clade (1 = ST19-L1, 2 = ST19-L2, 3 = ST19-L3, 4 = ST19-L4, 5 = ST313-L1, 6 = ST313-L2). Bar charts show the number of isolates per clade coloured by the country of isolation. **c** Distribution of *S*. Typhimurium isolates over time assigned to an invasive *S*. Typhimurium clade (1 = ST19-L1, 2 = ST19-L2, 3 = ST19-L3, 4 = ST19-L4, 5 = ST313-L1, 6 = ST313-L2). Historical isolates (older than 1975) were excluded in this representation.

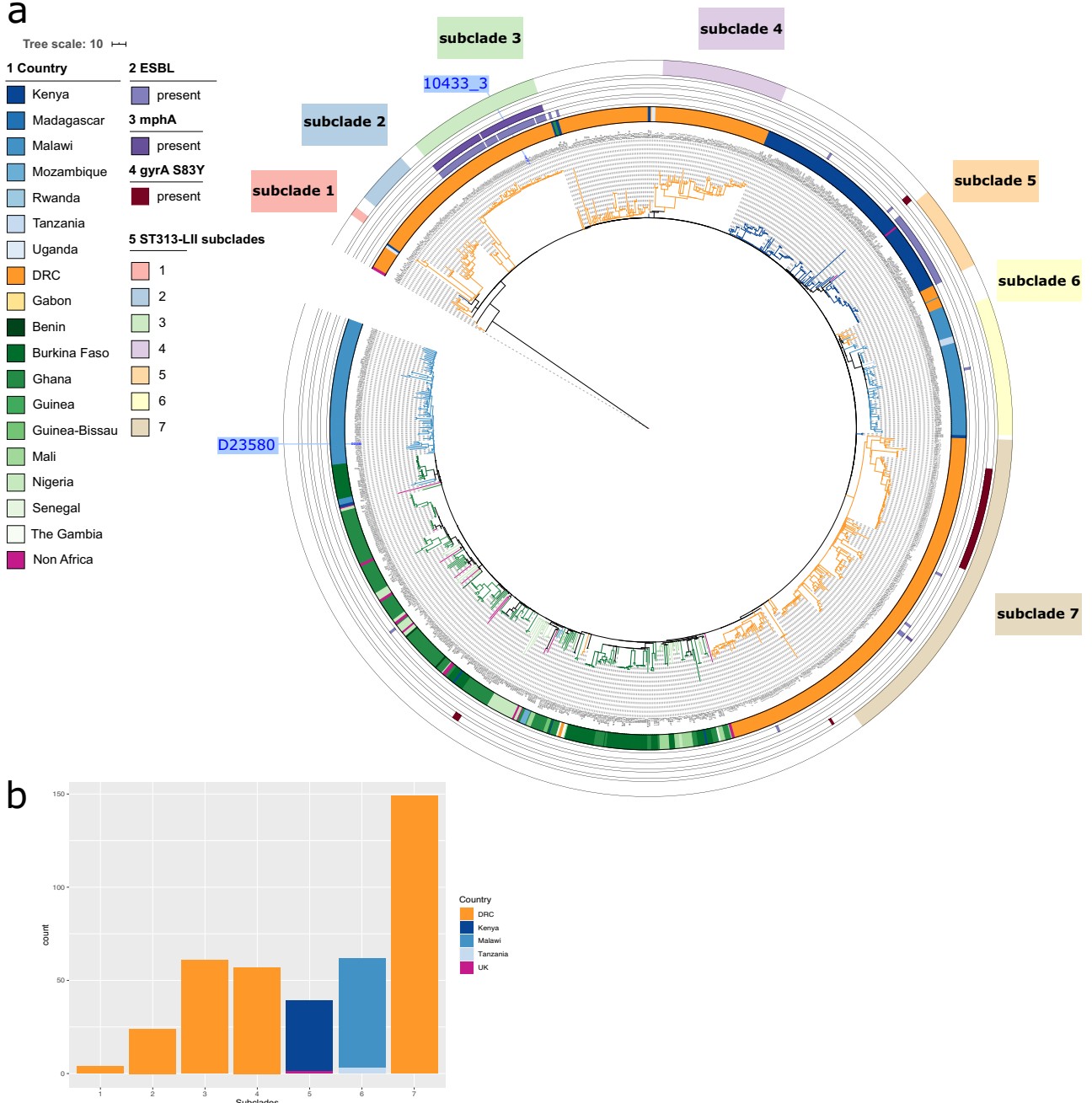

**Fig. 2 | The distribution of invasive *S.* Typhimurium clade ST313-L2 in Africa.** **a** Maximum likelihood phylogenetic tree of ST313-L2 isolates, based on mapping to reference strain D23580 (highlighted in blue in the phylogeny). The tree is based on 5380 chromosomal SNPs. The tree is rooted with *S.* Typhimurium strain DT2B, a European ST313 strain. Branches are coloured by the country of isolation. ST313-L2 subclades as identified in this study are coloured in the outher circle. Subclade 3 confers with the previously called ST313 sublineage II.1, and reference isolate 10433_3 is highlighted in blue[10]. Metadata are visualised on the concentric rings in compliance to the legend, from the inside to outside; (1) Country of origin, (2–4) antimicrobial resistance markers associated with XDR and PDR outbreaks (ESBL, azithromycin resistance and the GyrA S83Y mutation associated with decreased susceptibility to ciprofloxacin). (5) ST313-L2 subclades. Branch lengths represent the number of SNPs as indicated in the scale bar. **b** Distribution ST313-L2 subclades per country. Bar charts show the number of isolates per subclade coloured by the country of isolation.

markers, which was either a *qnr* acquisition, chromosomal GyrA D87N or GyrB S464Y substitutions. Strikingly, XDR or PDR isolates always harboured an acquired plasmid: XDR coincided with either the presence of IncHI1 ($n = 4$) or IncHI2 ($n = 50$) replicons, while PDR was associated with the presence of IncHI2 plasmids ($n = 3$). The IncHI2 plasmid conserved in ST313-L2 subclade 3 carries the XDR resistance markers and was found to have high similarity with the IncHI2 plasmids of ST313-L2 isolates from Kenya and Malawi[10].

We resolved the full plasmid sequence of all available XDR and PDR isolates and subjected them to a comparative analysis (overview in Table 2). IncHI2 plasmids contributing to XDR and PDR in invasive *S.* Typhimurium isolates in sSA were highly related to each other, but also to non-XDR/PDR IncHI2 plasmids (Fig. 6a). High similarity was, for example, observed with a plasmid from a non-invasive isolate from Morocco (S15BD01306) carrying mobilised colistin resistance gene (*mcr-9*). IncI1 plasmids contributing to XDR in invasive *S.* Typhimurium

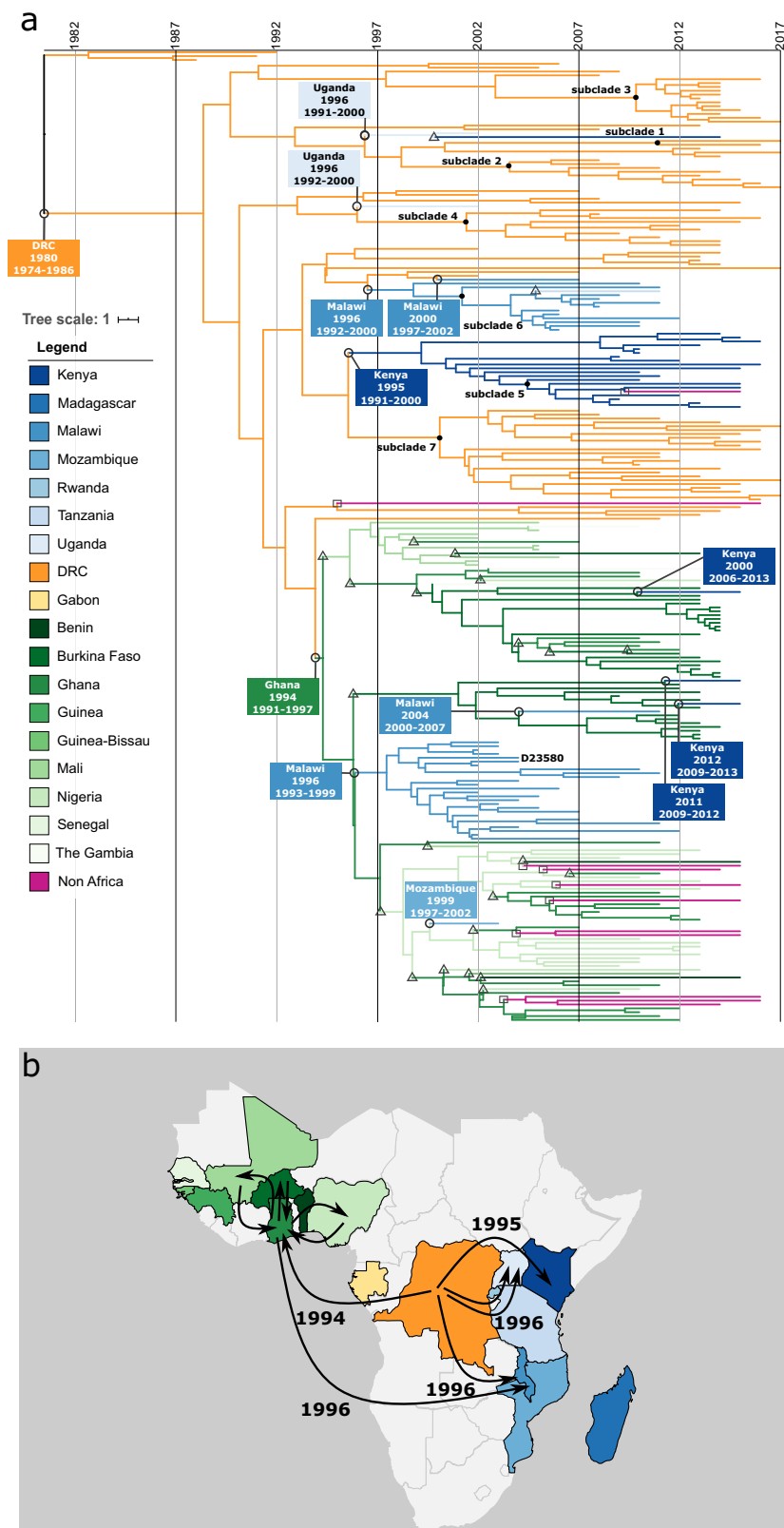

isolates in sSA were similarly highly related to each other, but also to other IncI1 plasmids (Fig. 6b).

All XDR and PDR isolates originate from the DRC, and were found in multiple provinces (Sankuru, Tshopo [Kisangani], Kongo-Central [Kisantu]), suggesting a widespread of the resistance plasmids in the country. Related IncI1 and IncHI2 plasmids were however identified across Africa, i.e., respectively in Nigeria and Kenya, and in Kenya, Malawi and Morocco, which suggests a possible widespread reservoir of these types of plasmids and thus AMR across the continent. Strikingly, the historical isolate AA00491 (DRC, 1984) carried a hybrid IncI1-IncHI2 plasmid showing similarity to both the recent IncI1 and IncHI2 plasmids.

**Fig. 3 | Estimated spread of ST313-L2 across sub-Saharan Africa. a** Time-tree from BEAST showing phylogeographical reconstruction of ST313-L2. Estimated ages of nodes where transmission between African regions (East, Central and West) occurred are annotated with black circles and predicted years are reported with the 95% HPD interval. Triangles and squares indicate transmission events respectively within the region to a neighbouring country, and outside the continent (travel-associated). Branches and nodes are coloured according to the country with the highest posterior probability. Branch lengths represent the number of years as indicated in the scale bar. Isolate D23580 from Malawi is annotated in the three, as well as all ST313-L2 subclades (black circle at most recent common ancestor). Subclade 3 coincides with the previously identified ST313 sublineage II.1[10]. Supplementary Fig. 8 presents the raw data of (**a**) with confidence intervals included. **b** A map showing the transmission events between the African regions (East, Central and West) and the respective predicted years of transmission. Additional cross-country transmission is observed in West Africa after the introduction in the region in 1994, annotated with arrows in the region.

## Table 1 | Quinolone resistance-determining region (QRDR) SNP acquisitions in African invasive *S.* Typhimurium clades

| QRDR event | N | Location | Year | Amino acid substitution | Clade |
|---|---|---|---|---|---|
| 1 | 3 | Kenya, Nairobi | 2005, 2014, 2014 | GyrA D87Y | ST313-L1 |
| 2 | 1 | DRC, Bwamanda | 2008 | GyrA D87Y | ST313-L2 |
| 3 | 1 | DRC, Kisangani | 2011 | GyrA D87N | ST313-L2 subclade 2 |
| 4 | 1 | DRC, Kinshasa | 2008 | GyrA D87N | ST313-L2 subclade 3 |
| 5 | 4 | Kenya, Siaya | 2009, 2010, 2011, 2012 | GyrA D87G | ST313-L2 |
| 6 | 3 | Kenya, Kombewa | 2010, 2013 | GyrA S83Y | ST313-L2 |
| 7 | 1 | DRC, Lwiro | 2007 | GyrA S83F | ST313-L2 |
| 8 | 1 | DRC, Lwiro | 2002 | GyrA S83F | ST313-L2 |
| 9 | 1 | DRC, Kisangani | 2015 | GyrA D87Y | ST313-L2 |
| 10[a] | 49 | DRC, Kisantu | 2013–2017 | GyrA S83Y | ST313-L2 subclade 7 |
| 11 | 3 | DRC, Kisantu | 2017 | GyrA D87N | ST313-L2 subclade 7 |
| 12 | 1 | DRC, Kisantu | 2014 | GyrA S83Y | ST313-L2 |
| 13 | 1 | DRC, Kisantu | 2014 | GyrA S83F | ST313-L2 |
| 14 | 2 | DRC, Kisantu | 2009, 2012 | GyrA D87Y | ST313-L2 |
| 15 | 3 | Ghana | 2010, 2012 | GyrB E466D | ST313-L2 |
| 16 | 3 | Ghana | 2011 | GyrA S83Y | ST313-L2 |
| 17 | 1 | Benin | 2015/2016 | GyrA D87Y | ST313-L2 |

Isolates with QRDR SNPs are listed. When isolates formed a monophyletic group showing the same genetic marker, they were grouped as a single QRDR SNP acquisition event. For each isolate or group of isolates their location and year of isolation is given, as well as the resulting amino acid substitution resulting from the QRDR SNP and the invasive *S.* Typhimurium clade as defined in this study.

[a]The 49 isolates within ST313-L2 subclade 7 are not listed separately.

## Hotspots for AMR in *S.* Typhimurium are found in Kenya and DRC

PDR and XDR previously had only been detected in DRC while ESBL production and QRDR SNP acquisitions were observed across sSA. PDR was observed in all three sites in DRC with extensive bloodstream surveillance; Kinshasa, Kisangani and Kisantu in the DRC. However, in Kisangani, Kisantu, Kinshasa, as well as in Nairobi (Kenya), isolates with XDR or ESBL-producing plasmids co-circulated with isolates showing QRDR SNPs, presenting a risk for PDR through plasmid transfer (Fig. 7).

Most frequently, isolates with increased AMR presented as single, sporadic cases, although in the Kisantu hospital area in the DRC, both an XDR outbreak (ST313-L2 subclade 3) and a clade with a QRDR SNP (ST313-L2 subclade 7) were observed. These outbreaks of ST313-L2 subclades 3 and 7 underly two separate waves with increased cases, respectively from 2013 Q4 to 2015 Q1 and 2016 Q1 to 2017 Q3 (Supplementary Fig. 11).

## Evolutionary processes in invasive *S.* Typhimurium clades

Invasive ST313 isolates have previously been associated with signatures of genome degradation associated with host adaptation[17]. To assess this here, non-synonymous SNPs were extracted, and the genes potentially inactivated in the invasive *S.* Typhimurium clades were analysed (Supplementary Table 1). Twenty-six such genes were identified in at least two invasive *S.* Typhimurium clades, and thus present parallel evolution towards loss of these genes in the separate invasive *S.* Typhimurium clades. Three of these genes were found in three of the six invasive *S.* Typhimurium clades, i.e., *malZ* (ST19-L2, ST19-L3 and ST313-L1), *ssrA* (ST19-L3, ST313-L1 and ST313-L2) and *stfC* (ST19-L4, ST313-L1 and ST313-L2). These genes are involved in virulence in a gut or macrophage environment[18–22] or survival outside the host[23]. Twenty-three genes showed non-synonymous SNPs in two of the six invasive *S.* Typhimurium clades and were related to metabolism (*pdxK*, *eutA*), host interactions/virulence (*shdA*, *steC*), membrane/surface-associated processes (*dacD*, *fimA*) redox-associated processes (*ydiJ*, *ccmH2*, *bcp*, *SL1344_1475*) or phage (*SL1344_2600*, *SL1344_2582*). Of these, none have previously been found as pseudogenes in human-associated *S.* Typhimurium. However, *fimA* has been found as a pseudogene in *Bordetella pertussis*[24], *ydiD* and *shdA* are known pseudogenes in *S.* Typhi and other host-adapted serovars[25–27], and *steC* is a pseudogene in some passerine-adapted *S.* Typhimurium[28,29]. Other molecular mechanisms, such as subtle changes in expression of expression pumps might also underly the observed differences in MIC values.

When analysed within a global *S.* Typhimurium context, the six invasive *S.* Typhimurium clades from sSA presented different characteristics than global *S.* Typhimurium isolates, which would suggest that they have spread differently and have undergone different levels of host adaptation (Fig. 8). Overall, the invasive *S.* Typhimurium clades form distinct clades in the global phylogeny. Their difference from global *S.* Typhimurium isolates underlines their specific importance for sSA rather than across the globe, with however, few global isolates interspersed with ST19-L1 and ST19-L4 isolates.

We further investigated the genome degradation of lineages using the DBS, which is a measure for *Salmonella* invasiveness based on the content of pseudogenes in the isolate's genomes. *S.* Typhimurium

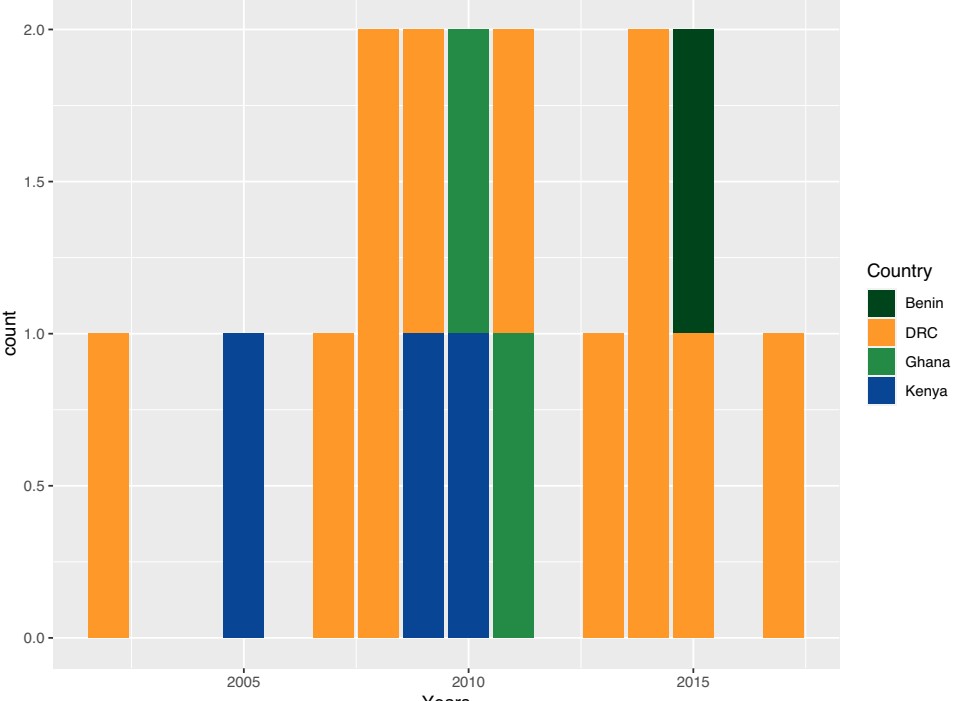

**Fig. 4 | Continuous emergence of individual genetic events of quinolone resistance-determining regions (QRDR) SNP acquisitions in African invasive *S.* Typhimurium clades over time.** Events are plotted per year and coloured by the country of isolation. The years of the oldest isolate from a genetically related cluster of isolates presenting the same QRDR SNP are plotted. DRC Democratic Republic of Congo.

lineages belonging to global clade-α[30] had a low DBS, indicative for little genome degradation and lower signatures of invasiveness, and included ST19-L1 and ST19-L2 with DBSs of −0.0089 and 0.07, respectively (Fig. 8b). The lack of genome degradation in ST19-L1 was consistent with this, containing strains with a broad host range. In contrast, ST19-L2 had an elevated DBS, suggesting greater host adaptation, although there might be sampling bias or a local outbreak underlying these data as the ST19-L2 isolates were sampled over a short time frame at only one location. *S.* Typhimurium lineages belonging to global clade-β[30] had a wider range of DBSs. ST19-L3 and ST19-L4 (both ST19) had a mean DBS of 0.0337 and 0.0013, respectively. The high DBS of ST19-L3 is consistent with a highly degraded genome, considered to be a signature for increased invasiveness and similar to that described in the neighbouring DT56 complex. ST19-L4 has emerged from the DT204 complex that caused an MDR epidemic in Europe during the 1980s[31]. ST313-L1 and ST313-L2 are part of clade-β and present moderate to high genome degradation with mean DBSs of 0.0105 and 0.0197.

## Discussion

We present an overview of invasive *S.* Typhimurium in sSA, maximally using a geographically and temporally balanced dataset of 1302 African isolates. Okoro et al. previously identified two phylogenetic ST313 lineages (called 1 and 2) as drivers of the *S.* Typhimurium epidemic in sSA[9]. We identified four further significant clades and underlined the importance of ST313-L2 in the current pandemic. Our data identified similar evolutionary signatures in each clade involving the acquisition of multiple AMR determinants and signs of genome degradation, potentially linked to host adaptation. Two of the ST19 clades were observed recently to be dominant in Kenya and were called the Kenyan ST19 lineages I and II[11], coinciding with ST19-L3 and ST19-L4. The other two ST19 clades described here, have not been described earlier. Another recent study has identified a novel clade in Malawi, likely presenting a seventh emergence of an invasive *S.* Typhimurium clade[16].

We present an updated numbering system to annotate those different invasive *S.* Typhimurium lineages with ST313-L2 as the current dominant clade in sSA, associated with highest levels of AMR. The emergence of multiple, independent lineages across the *S.* Typhimurium population, presenting similar evolutionary patterns towards invasiveness, implies that *S.* Typhimurium, ST19 and ST313, have the intrinsic capability of becoming invasive. In addition, as long as these drivers are present, one can assume that it is likely that new invasive lineages will continue to appear.

A limitation to this and other iNTS genomics studies is the scarcity of bacterial isolates available from bloodstream infections in sSA, associated with intrinsic sampling bias. First, healthcare utilisation is relatively low and frequently delayed in sSA. Second, blood culture analysis and surveillance are increasingly performed across sSA but are still not routinely done in all countries as it is expensive and poorly implemented in clinical care. Lastly, sample storage and sequencing are not systematically done on all samples. Whilst this situation is changing, this study is therefore opportunistic but nonetheless covers multiple regions across sSA, combining data from blood culture surveillance projects undertaken across sSA in the recent decades. The intrinsic sample bias however implies that patterns, such as the diversity of clades, that we observed between regions might be skewed. We have however taken steps throughout the study to mitigate these effects, including the use of rigourous statistical methods, to present the best possible predictions.

We postulate that the current pandemic ST313-L2 clade emerged in the DRC in 1980 and has spread across sSA from the DRC. Historical invasive *S.* Typhimurium isolates also have only been observed and reported in DRC and Rwanda. Although this might be a causal effect of the clade only being present in Central Africa, sample scarcity and bias could be a limitation for this analysis.

The success of ST313-L2 has earlier been contributed to its chloramphenicol resistance[9]. Chloramphenicol was used frequently in DRC as first choice treatment for bacterial meningitis and typhoid

**Fig. 5 | Observed molecular mechanisms of extensive drug resistance (XDR) and pan-drug resistance (PDR) in invasive *S*. Typhimurium.** XDR in invasive *S*. Typhimurium presented as either caused by a plasmid carrying genetic markers for extended-spectrum beta-lactamase (ESBL) activity and azithromycin (AZI) resistance or by a plasmid carrying genetic markers for ESBL and fluoroquinolone (FQ) resistance. PDR in invasive *S*. Typhimurium presented as either caused by a plasmid carrying XDR in an isolate carrying a SNP in a quinolone resistance-determining region (QRDR) or by a plasmid carrying resistance markers for ESBL, AZI and FQ. The plasmid type is annotated per the observed mechanism. The specific plasmids per isolate are listed in Table 2. *Described as part of ref. 10.

fever[32], and we observe chloramphenicol resistance in ST19-L4 from Lwiro, DRC isolated in 1984–1985. The main transmission events of ST313-L2 at the continental level happened in a time frame of only 2 years. Coincidently, major wars took place in de DRC during this period being the First Congo War or Africa's First World War (1996–1997) and the Second Congo War or the Great African War (1998–2003) took place. During that time, there was substantial travel due to important foreign involvement and there were large numbers of refugees in the DRC after the Rwandese War (1994)[33]. At the same time, the HIV pandemic was still expanding in sSA, creating a large susceptible population to iNTS infections[9]. Political instability may, therefore, have played a role in the intracontinental spread of invasive *S*. Typhimurium, which has also been seen for other diseases such as cholera[34].

Higher-order subclades present in ST313-L2 were confined to specific countries, suggesting a slow local spread of iNTS at the country level. Two subclades with increased AMR have emerged in DRC, namely one presenting XDR and one DCS, again implying that DRC might be a source for emerging subpopulations. Similarly, one ESBL ST313-L2 sub-branch emerged from Kenya, a country also presenting a high diversity of invasive *S*. Typhimurium clades and suggesting that Kenya might be another hotspot for emerging

subpopulations. Four specific locations were identified as hotspots where IncHI2-plasmid-driven resistance coincides with chromosomal DCS (Kisantu, Kisangani and Kinshasa in DRC; Nairobi in Kenya). It will be important in the coming years to maintain genomic surveillance in these sites, and others. At the moment, single PDR cases have been reported in DRC[10]. Acquisition of XDR and PDR involved plasmids highly related to the IncHI2 and IncI1 plasmids, suggesting the existence of a mobile AMR-reservoir for *S*. Typhimurium in sSA. These IncHI2 and IncI1 plasmids are commonly found in the *S. enterica* species[35], but also in other related organisms such as *Escherichia coli*[36], with which the invasive *S*. Typhimurium isolates might exchange plasmids.

The increase of AMR is worrying in *S*. Typhimurium, especially with limited antimicrobials available and resistance being observed against all available antimicrobials. iNTS vaccines are currently in development[37], but it is however unlikely for these to become licensed and WHO prequalified before the end of the 2020 s. In addition, specific risk factors for iNTS infection (malnutrition, HIV and malaria coinfections) still are significant in the human population. There has never been a clinical endpoint trial of the management of iNTS disease and there is an urgent need for holistic approaches to prevention

**Table 2 | IncI1 and IncHI2 plasmids associated with antimicrobial resistance (AMR) identified in African invasive S. Typhimurium isolates**

| Isolate | invasive S. Typhimurium clade | Plasmid | Length | Resistance of isolate | Type | Origin | Resistance genes on plasmid | Reference |
|---|---|---|---|---|---|---|---|---|
| 10530_17 | ST313-L2 | pSTM-10530_17 | 296484 bp—circular | XDR[a] | IncHI2 | DRC Sankuru, 2017 | blaTEM, catA, sul, dfrA, blaCTX-M-15, blaOXA-1, aac(6')-Ib-cr, qnrB | This study |
| 3152_4 | ST313-L2 | pSTM-3152_4 | 262211 bp—circular | MDR | IncHI2 | DRC Kisangani, 2014 | Aac(3)-Id, aph(3')-Ia, catA1, tetB | This study |
| 5390_4 | ST313-L2 subclade 7 | pSTM-5390_4 | 281440 bp—circular | PDR | IncHI2 | DRC Kisangani, 2016 | blaTEM, sul, dfrA, blaSHV-12, mphA, aac(6')-Ib-cr, qnrS | This study |
| 7593_12 | ST313-L2 subclade 7 | pSTM-7593_12 | 267916—circular | XDR | IncHI2 | DRC, Maniema, 2012 | aadA1, blaSHV-12, blaTEM-1B, catA, qnrS, sul1, tetB, dfrA1 | This study |
| ST313-L2 subclade 3 (10433_3) | ST313-L2 subclade 3 | pSTM-ST313-II.1 | 274695 bp—circular | XDR (PDR)[a] | IncHI2 | DRC Kisantu, 2014 | blaTEM, catA, sul, dfrA, blaSHV-12, mphA | 10 |
| ST313-L2 subclade 4 | ST313-L2 subclade 4 | pKST313 | Circular | MDR + ESBL | IncHI2 | Kenya, 2009 | Aac(3)IIa, aac(6')-Ib-cr, aadA1, aph(3'')-Ib, aph(6)-Id, blaCTX-M-15, blaOXA-1, blaTEM-1B, catA, catB, aac(6')-Ib-cr, sul2, tetA, dfrA14 | 13 |
| A54560 | ST313-L2 subclade 6 | pSTM-A54560 | Circular | MDR + ESBL + FQ (XDR)[d] | IncHI2 | Malawi, 2009 | Aac(3)IIa, aac(6')-Ib-cr, aadA1, aph(3'')-Ib, aph(6)-Id, blaCTX-M-15, blaOXA-1, blaTEM-1B, catA, catB, aac(6')-Ib-cr, qnrB, sul2, tetA, dfrA14 | 43 |
| S15BD01306 | ST313-L2[b] | pSTM-S15BD01306 | Contigs | XDR[c] | IncHI2 | Morocco, 2015 | aadA2b, mcr-9, sul1, dfrA16 | This study |
| AAO0491 | ST19-L4 | pSTM-AAO0491 | Contigs | MDR | IncHI2 | DRC Kinshasa, 1984 | – | This study |
| 23060_3 | ST313-L2 subclade 7 | pSTM-23060_3 | 97430 bp, circular | XDR | IncI1 | DRC Kisantu, 2017 | Sul, blaCTX-M-15, qnrS | This study |
| 22400_3 | ST313-L2 subclade 7 | pSTM-22400_3 | 90103 bp, circular | XDR | IncI1 | DRC Kisantu, 2017 | blaCTX-M-15, qnrS | This study |
| 22392_3 | ST313-L2 | pSTM-22392_3 | 90095 bp, circular | XDR | IncI1 | DRC Kisantu, 2017 | blaCTX-M-15, qnrS | This study |
| 8892 | ST313-L2 | pSTM-8892 | Contigs | MDR + ESBL | IncI1 | Kenya, Nairobi 2010 | blaCTX-M-15 | 13 |
| K171 | ST313-L2 | pSTM-K171 | Contigs | MDR + ESBL | IncI1 | Nigeria, Kano, 2012 | – | This study |
| AAO0491 | ST19-L4 | pSTM2-AAO0491 | Contigs | MDR + ESBL | IncI1 | DRC Kinshasa, 1984 | Aph(3')-IIa, aph(6)-Ic, blaTEM-1B | This study |

Invasive S. Typhimurium clades are included, based on the cluster analysis presented in this study. Resistance of the isolate is based on the genetic content of resistance makers (MDR = multidrug resistance, defined as co-resistance to chloramphenicol, ampicillin and co-trimoxazole; XDR = extensively drug resistance, defined as co-resistance of 2 of the 3 current treatment options (azithromycin (AZI), fluoroquinolone (FQ) and ceftriaxone) with MDR; PDR = pan-drug resistance, defined as co-resistance of AZI, FQ and ceftriaxone with MDR. Lengths are given for plasmid sequences resolved with long-read sequences, while 'contigs' is indicated when contigs from Illumina assemblies were used as input.
[a]Subclade 6.3 contains isolates with QRDR SNPs giving rise to PDR. The reference isolate 10433_3 for which the plasmid was resolved however does not carry the SNP.
[b]S15BD01306 was isolated from human stool and was not part of an invasive S. Typhimurium clade, but was included in the analysis for context.
[c]S15BD01306 carried the MCR resistance marker conferring resistance to colistin in addition to XDR markers. Colistin is not used for the treatment of invasive S. Typhimurium infections due to practical reasons (availability, price and administration).
[d]A54560 was previously not reported as XDR, but was here because of the presence of qnrB.

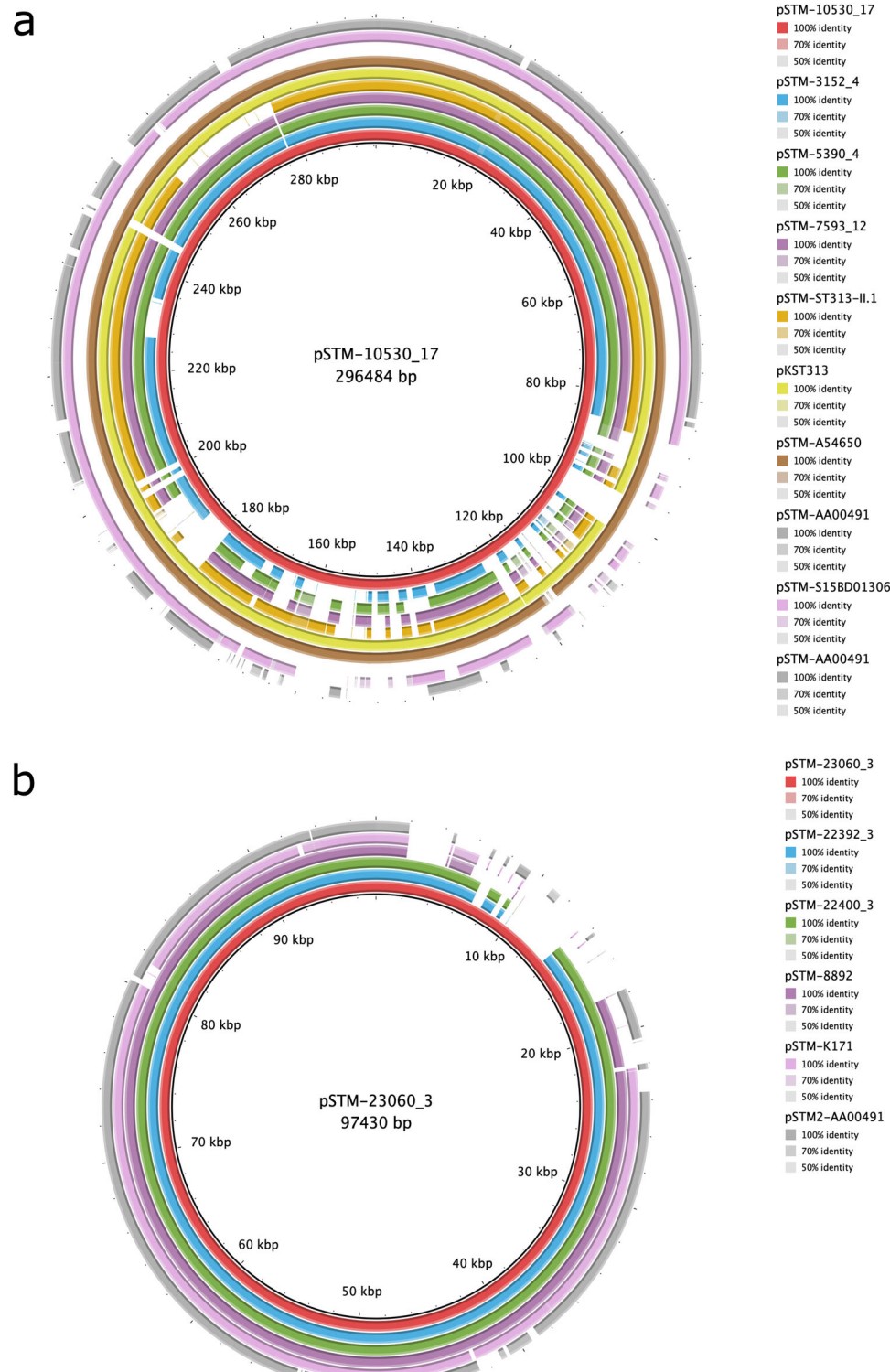

**Fig. 6 | Related IncHI2 and IncI1 plasmid drive XDR and PDR in African invasive** *S.* **Typhimurium.** Pairwise similarity of IncHI2 and IncI1 plasmids of invasive *S.* Typhimurium isolates. **a** Pairwise comparison of IncHI2 plasmid pSTM-10530_17 with pSTM-3152_4, pSTM-5390_4, pSTM-7593_12, pSTM-ST313-II.1[10], pKST313[13], pSTm-A54650[43] and Illumina assembled contigs of pSTM-S15BD01306 and pSTM-AA00491. **b** Pairwise comparison of IncI1 plasmid pSTM-23060_3 with pSTM-22392_3, pSTM-22400_3 and Illumina assembled contigs of pSTM-8892[13], pSTM-K171 and pSTM2-AA00491.

combining novel treatment strategies and preventive measures, including vaccine development and meeting Sustainable Development Goals around water, sanitation and hygiene (WASH), nutrition and control of malaria and HIV in endemic regions. Our study supports and urges for further action by providing a timely and continental perspective of invasive *S.* Typhimurium and its AMR in sSA.

## Methods
### Bacterial isolates
A total of 1419 *S.* Typhimurium isolates were included in this study, containing 1302 isolates from Africa. These originated from surveillance studies in 19 countries spanning different subregions of Africa (East, Central and West) (Supplementary Data 1 and Supplementary

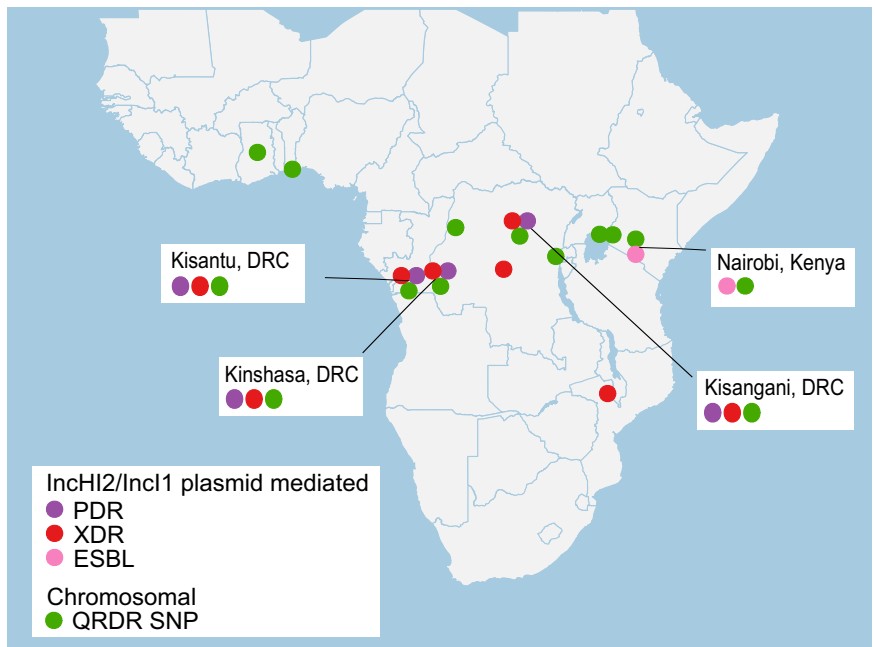

**Fig. 7 | Antimicrobial resistance (AMR) hotspots for invasive *S.* Typhimurium in sub-Saharan Africa.** GPS locations are plotted for isolates showing IncHI2/IncI1 plasmid-mediated pan-drug resistance (PDR), extensively drug resistance (XDR) and extended-spectrum beta-lactamase (ESBL) activity as well as locations with isolates presenting chromosomal quinolone resistance-determining region (QRDR) SNPs. Locations where plasmid-driven PDR, XDR or ESBL isolates co-circulate with chromosomal QRDR SNPs are annotated on the map, presenting a risk for increased AMR through plasmid transfer.

Fig. 1). Of these African isolates, 115 (8.1%) isolates were part of the study by Okoro[9] providing the *S.* Typhimurium overview in 2012, and including the ST313 lineage II D23580 isolate, used here as reference genome[38]. An additional 815 (57.4%) isolates were sequenced as part of the studies with isolates from sSA from Van Puyvelde et al.[10,39], Post et al.[40], Kariuki et al.[11,13,41], Feasey et al.[42], Msefula et al.[43], Park et al.[14] and Maclennan et al. (in preparation). As part of this study, 372 (26.2%) additional isolates from sSA were whole genome sequenced. Of these, three isolates originate from bacterial surveillance in the Centre National Hospitalier Universitaire Hubert Koutougou MAGA of Cotonou, Benin; 33 (2.3%) originate from bloodstream surveillance studies and human reservoir and transmission studies conducted in the Clinical Research Unit of Nanoro (CRUN) in Burkina Faso, 134 (9.4%) originate from the Democratic Republic of Congo (DRC), including ongoing bloodstream surveillance[44–47] and historical isolates which were bio-banked in the University Hospital Saint-Pierre Brussels (Belgium)[48–51]; 3 (0.2%) isolates originate from returning travellers from Gabon, Guinea and Morocco stored by Sciensano (Belgium); 86 (6.0%) isolates originate from the Malawi Liverpool Welcome (MLW) bacteraemia archive and dedicated stool samplings by the MLW Research Programme in Blantyre, Malawi[52]; 29 (2.0%) isolates from Nigeria originate from the Community Acquired Bacteremic Syndrome in Young Nigerian Children CABSYNC study and the Community Acquired Pneumonia and Invasive Bacterial Disease (CAPIBD), 69 (4.9%) from Rwanda of which one recent isolate collected by the Rwandan National Reference Laboratory in Kigali (Rwanda) and 68 historical isolates which were bio-banked in the University Hospital Saint-Pierre Brussels (Belgium)[48–51]; 15 (1.1%) isolates originate from population-based surveillance by the Medical Research Council (MRC) Unit The Gambia at the London School of Hygiene & Tropical Medicine (LSHTM) in the Basse region of The Gambia[53]. We have included the 75 (5.3%) published sequences from Ashton et al.[12], including ST313 sequences isolated by Public Health England in the UK, as context. Information on the sample source was available for 1102 of the African *S.* Typhimurium isolates, and include abattoir (1; 0.1%) human blood (1006; 91.4%),

stool (84; 7.6%), urine (4; 0.4%), pus (1; 0.1%), cerebrospinal fluid (4; 0.4%) and unspeccified human sample (1; 0.1%). Sixty-one of the 84 African stool isolates originate from endemic settings and presented invasive clones in the stool samples[11,54]. All available information on the isolates is available in Supplementary Data 1.

**Genome sequencing**

Each of the collaborating laboratories used their own individual methodologies for the extraction of genomic DNA. Index-tagged paired-end Illumina sequencing libraries were prepared as previously described[55]. These were combined into pools of 96 uniquely tagged libraries and sequenced on the Illumina HiSeq 2000 or HiSeq 2500 platform (San Diego, USA) according to the manufacturer's protocols to generate paired-end reads of 100–150 bp in length. Sequencing read quality was confirmed for all samples, including the presence of sequencing adapters, contamination level, GC fraction, insert size, qX yield, match with reference genomes and coverage of mapping, using the sequencing pipelines (Wellcome Sanger Institute, Hinxton, UK). Purity of the samples was confirmed by sorting reads per species for each sample using Kraken v.1.1.1[56]. Samples were assembled using Velvet v.1.2.10[57] and needed to cover at least 4.5 Mbp of the genome. Detailed statistics of the individual assemblies, including the number of contigs, average contig lenthg, max contig length and N50 are given in Supplementary Data 2.

Seven isolates from this study were subjected to MinION sequencing to resolve their plasmid sequences (Supplementary Data 3). These included all isolates which were available for additional sequencing and which showed high AMR (XDR, PDR and/or ESBL). For monophyletic branches with the same AMR profile and plasmid content, one isolate was subjected to MinION sequencing. DNA for MinION sequencing was extracted using the MasterPure Complete DNA and RNA Purification Kit (Epicentre, Madison, USA), following the manufacturer's guidelines. Nuclease-free water (Thermo Fisher scientific, Waltham, USA) was used to resuspend the DNA. Genomic DNA extracts were natively barcoded (with EXP-NBD104, SQK-LSK109) and

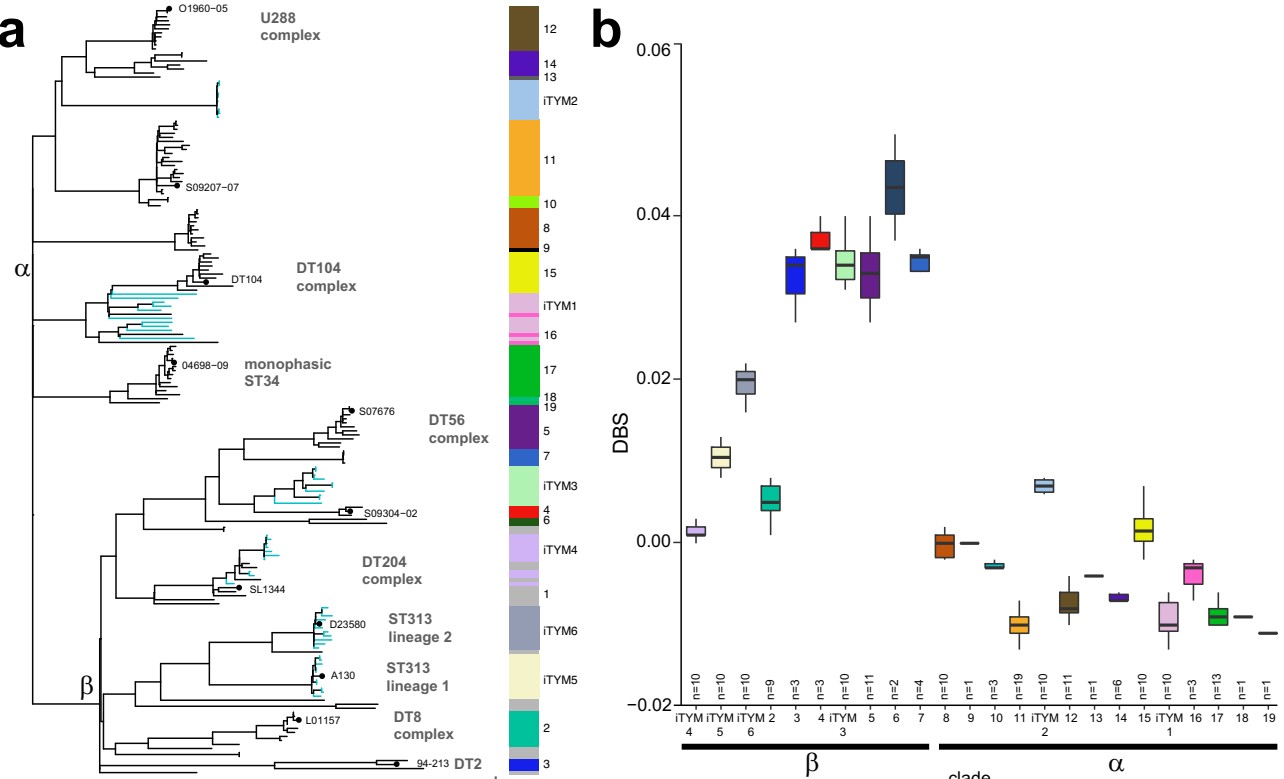

**Fig. 8 | Phylogenetic relationship of invasive *S.* Typhimurium clades in the global population structure of *S.* Typhimurium. a** Maximum likelihood phylogenetic tree containing 10 isolates from each invasive *S.* Typhimurium clade (blue branches) and a representative collection of 131 global *S.* Typhimurium isolates (black branches). The tree was constructed using sequence variation (SNPs) in the core genome with reference to *S.* Typhimurium strain SL1344. The root was identified using *S.* Heidelberg (accession number NC_011083.1) as the outgroup (not shown). The heatmap to the right of the tree highlights the invasive *S.* Typhimurium clades and the 19 previously determined population structure groups of the 131 strains (Bawn et al.,[30]). Isolate names are shown as branch labels where high-quality long-read reference sequences, clonal complexes described in literature are also shown. The bifurcations giving rise to *S.* Typhimurium clade-α and clade-β are annotated. **b** Box plots of the geometric mean Delta Bitscore (DBS: bitscore SL1344 (FQ312003)−test strain bitscore) of proteomes of the same representative collection of 131 *S.* Typhimurium isolates are shown. Centres represent the geometric mean DBS, minima and maxima of the boxes represent the first and third quartiles, vertical lines indicate first or second quartile + 1.5× the interquartile range, and outliers > or less than these values plotted as points. The number of genomes (biological replicates) analysed in each group are indicated within the figure (n = ). Boxes are coloured to indicate phylogroups in (**a**). Basally rooted population structure group 1 not shown.

sequenced on a MinION using flowcell type R9.4.1. The sequence was determined from the raw fast5 output files using guppy basecaller in high accuracy mode (default parameters with --config dna_r9.4.1_450bps_hac.cfg, Guppy basecalling suite, (C) Oxford Nanopore Technologies, Limited. v3.0.3) to obtain fastq files. These fastq files were demultiplexed with guppy barcoder v3.0.3 and porechop v0.2.3 (default parameters with --barcode_kit EXP-NBD104). Hybrid assemblies from Nanopore and Illumina read sequences were constructed using Unicycler v0.4.6[58] and the assembly graphs were evaluated using Bandage[59]. Sequences were then annotated using PROKKA v1.11[60].

### In silico identification of AMR and plasmid replicons

To detect AMR genes, the ARIBA software v.2.14.6[61] with default thresholds and CARD database v.3.0.7[62] were used. For the detection of plasmids, the PlasmidFinder[63] database was used with the mapping-based allele tool SRST2 v.0.2.0[64].

MDR was defined as the co-presence of genetic markers conferring resistance to chloramphenicol, ampicillin and co-trimoxazole; XDR as the co-presence of genetic markers which are responsible for the resistance to two of the three current treatment options (AZI, FQ or ceftriaxone) on top of MDR; pan-drug resistance (PDR) was defined as co-presence of genetic markers conferring resistance to AZI, FQ and ceftriaxone combined with MDR[10,15].

### SNP analysis

Illumina HiSeq reads were mapped to the *S.* Typhimurium reference genomes of ST313 lineage II (D23580, FN424405.1[38]) using SMALT v0.7.4 to produce a BAM file. SMALT was used to index the reference using a kmer size of 20 and a step size of 13, and the reads were aligned using default parameters but with the maximum insert size set as three times the mean fragment size of the sequencing library. PCR duplicate reads were identified using Picard v1.92 (Broad Institute, Cambridge, MA, USA) and flagged as duplicates in the BAM file.

Variation detection was performed using samtools mpileup v0.1.19 with parameters "-d 1000 -DSugBf" and bcftools v0.1.19[65] to produce a BCF file of all variant sites. The option to call genotypes at variant sites was passed to the bcftools call. All bases were filtered to remove those with uncertainty in the base call. The bcftools variant quality score was required to be greater than 50 and mapping quality greater than 30. If not all reads gave the same base call, the allele frequency, as calculated by bcftools, was required to be either 0 for bases called the same as the reference, or 1 for bases called as a SNP. The majority base call was required to be present in at least 75% of reads mapping at the base, and the minimum mapping depth required was 4 reads, at least two of which had to map to each strand. Finally, strand_bias was required to be less than 0.001, map_bias less than 0.001 and tail_bias less than 0.001. If any of these filters were not met, the base was called as uncertain.

## Phylogenetic analysis

A pseudo-genome was constructed by substituting the base call at each site (variant and non-variant) in the BCF file into the reference genome and any site called as uncertain was substituted with an N. Insertions with respect to the reference genome were ignored and deletions with respect to the reference genome were filled with N's in the pseudo-genome to keep it aligned and the same length as the reference genome used for read mapping.

Recombinant regions in the chromosome such as prophage regions and the *fljB* ORF in the chromosome were removed from the alignment and checked using Gubbins v1.4.10[66]. SNP sites were extracted from the alignment using snp-sites v.2.5.1[67] and used to construct a maximum likelihood phylogeny with RAxML v8.2.8[68] with substitution model GTR-GAMMA. For the full tree, support for nodes on the trees was assessed using 100 bootstrap replicates and *S.* Paratyphi A270 was included as an outgroup to root the tree. For the ST313-L2 tree, 1000 bootstrap replicates were used and isolate *S.* Typhimurium DT2 was included as an outgroup to root the tree. Trees were visualised using Figtree v1.4.2 and iTOL[69].

## Identification of clades and subclades

For the identification of clades and subclades within the African *S.* Typhimurium population, a similar approach was used as previously employed for the classification of *S.* Typhi[70]. FastBaps v.1.0.3[71] was used with the 'baps' prior. Hierarchal clustering analysis was run on the full dataset allowing identification of six clades, here defined as invasive Typhimurium clades ST19-L1–4 and ST313-L1-2. Clade ST313-L2 isolates (formerly defined as ST313 sublineage II) were separately subjected to hierarchal clustering thereby yielding seven subclades (ST313-L2 subclades 1–7). Hereto, ten nested levels of molecular variation were fitted to the data and level 2 allowed identification clusters until single-member clusters thereby following the approach used for *S.* Typhi classification[72], and the level 2 classification was further used for the (sub)clade definitions. The hierarchal clustering method is a phylogeny-free approach, and population structure was taken into account by integrating level 2 clusters with the phylogenetic trees.

Clades were defined as FastBaps clusters presenting as monophyletic branches. Neighbouring and nested clusters were merged in one clade while the differentiating branch showed a minimum of 50 SNPs difference. Neighbouring clusters were merged until the next cluster was dominated by non-African non-invasive isolates, allowing a representative classification of invasive African *S.* Typhimurium isolates in 6 major clades. This results in 21 of the 118 non-African isolates being clustered within the invasive *S.* Typhimurium clades. Unclassified isolates were assigned to the ancestry clade (Supplementary Data 1).

Subclades of ST313-L2 were defined similarly as clades, i.e., as FastBaps clusters presenting as monophyletic branches. All neighbouring and nested clusters were merged in subclades, with the differentiating branch showing at least 4 SNPs difference. Merging of neighbouring clusters was continued as far as possible. To maintain compatibility with the phylogeny, few single nested isolates needed to be reassigned. Some isolates did not reach the criteria for clustering as they were part of polyphyletic lineages and were assigned to the ancestry subclade of ST313-L2 This approach uses stringent settings, thereby limiting classification of all potential substructures of ST313-L2 but provides a framework with significant subclades and a nomenclature which can be extended in the future.

## Evolutionary context analysis

To determine the evolutionary context of the African strains, 10 representatives randomly selected from the phylogeny (Supplementary Data 4) of each invasive *S.* Typhimurium clade were placed in a phylogenetic context with strains from a well-characterised dataset of 131 *S.* Typhimurium strains[30] (Supplementary Data 5). Maximum

likelihood phylogenetic trees were constructed as previously described[30]. Briefly, reads were mapped to the SL1344 whole genome sequence assembly (FQ312003) and SNPs were determined using the Rapid haploid variant calling and core SNP phylogeny pipeline SNIPPY v.3.0 (https://github.com/tseemann/snippy). The software was also used to construct a sequence alignment of core genome variant sites from which a maximum likelihood phylogenetic tree was generated using the GTRCAT model implemented with an extended majority-rule consensus tree criterion in RAxML[73]. The Delta Bitscore (DBS) was calculated for the proteome inferred from each genome to estimate the level of genome degradation as previously described[30]. Briefly, reads were mapped to the SL1344 reference genome to create alignment sequences which were then annotated using Prokka v1.11[60]. Genes in each annotated sequence were then analysed in a pairwise fashion against SL1344 using DBS[74] as described previously[30]. The mean DBS per genome was plotted.

## Phylogeography

A representative selection of ST313-L2 isolates was made by first stratifying the dataset by year and using CD-HIT v.4.8.1[75] at a 0.99 threshold per group of isolates to identify isolates representing the genetic diversity within the group. A total of 220 isolates were selected using this approach and complemented with 31 isolates which were selected based on the RAxML phylogeny and present additional geographical transmission events in the maximum likelihood tree. Reference isolate D23580 was also included.

A temporal signal was identified in the dataset with TempEST v.1.5.3[76]. The phylogeography was reconstructed using BEAST v1.8.4[77]. BEAUti xml's were constructed to compare the different substitution models (GTR + I, GTR+Gamma), molecular clocks (Strict, Relaxed Log-Normal) and population size models (Constant, Exponential, Logistic, Expansion, Skygrid, Sky Ride, Bayesian Skyline, Extended Bayesian Skyline). The ratio of the marginal likelihoods were compared using Bayes factor[78,79] and yielded the general time reversible (GTR) substitution model with diffuse gamma distribution prior (shape 0.001, scale 1000) and invariant sites, an uncorrelated log-normal relaxed molecular clock and a model with exponential population size as optimal. Countries were added as traits in the BEAST model, and ancestors states were reconstructed at all ancestors.

Three independent runs of 80 million MCMC generations were calculated and samples were taken every 8000 generations. Log files were inspected in Tracer v1.7.1[80] for convergence, proper mixing, and sufficient sampling. A 3.75% burn-in was removed from each run. Log and tree files were combined using LogCombiner v2.5.0. The posterior sample of the time-trees were summarised in TreeAnnotator v1.8.4 to produce a maximum clade credibility tree with the posterior estimates of node heights visualised on it.

We inferred transmission events along branches where the predicted country changed between internal nodes in the final BEAST maximum credibility tree. These data were used to generate the inferred transmissions shown on the map. For the phylogeographical analysis, the Bayes_factor_comparison.py, replace_BEAST_blocks.py and prepare_BEAST_alignment.py scripts were used, available at Github page https://github.com/sanger-pathogens/bact-gen-scripts.

## Plasmid comparisons

XDR and PDR IncI1 and IncHI2 plasmids were pairwise compared using BRIG v.0.95[81]. The respective IncHI2 and IncI1 plasmids that were complete, circular and had the longest sequence were used as references for comparison, which were IncHI2 plasmid pSTM-10530_17 (296484 bp) and IncI1 plasmid pSTM-23060_3 (97430 bp). Included plasmid sequences obtained through this study using MinION sequences were IncHI2 plasmids pSTM-10530_17, pSTM-3152_4, pSTM-5390_4, pSTM7593_12 and IncI1 plasmids pSTM-23060_3, pSTM-22392_3, pSTM-22400_3, pSTM-8892. Public sequences of IncHI2

plasmids pSTm-ST313-II.1 (*S.* Typhimurium, ERS1310131), pKST313 (*S.* Typhimurium, LN794248) and pSTm-A54650 (*S.* Typhimurium, LK056646) were included.

A de novo assembly was performed based on Illumina sequences of isolates AA00491, S15BD01306, K171 and 8892[82] and the contigs of the plasmid sequence were identified using a pairwise blastN v. 2.10.0 analysis of these sequences with respectively pSTM-10530_17 and pSTM_23060_3 as the query. All contigs covering more than 3% of the query sequence were retained for this analysis. Contigs were reordered using Mauve version 2015_02_25[83].

**Ciprofloxacin susceptibility testing and statistical analysis**
Antimicrobial susceptibility of the DRC isolates was done before[47], following CLSI guidelines by assessing Minimum Inhibitory Concentrations (MIC) with the E-test macromethod (bioMérieux, Oxoid) for ciprofloxacin. An increase of MIC values per nested substructure of *S.* Typhimurium ST313-L2 subclade 7 was statistically analysed using a generalised linear model (glm function) using R software, with a Gaussian distribution. The nested substructures are given in Supplementary Fig. 2a.

**Ethics statement**
Ethical approval for the Microbiological Surveillance was granted by the Institutional Review Board of the Institute of Tropical Medicine, Anwerp (ref. 613/08, 23/03/2021 and ref. 1108/16, 23/08/2016), by the Ethics Committees of the Antwerp University Hospital, Belgium (ref. 08172613, 01/04/2021 and ref. 16/34/347, 09/01/2017) and the School of Public Health in Kinshasa, DRC (ref. ESP/CE/092/2021, 12/05/2021), University of Malawi College of Medicin Research Ethics Committee (COMREC P.06/20/3071), the Rwanda National Ethics Committee (ref. 903/RNEC/2018, 17/12/2018), the Comité Local d'Ethique Pour la Recherche Biomédicale de Parakou, Benin (0195/CLERB-UP/P/SP/R/SA), the Gambia Government/Medical Research Council Unit The Gambia Joint Ethics Committee (ref. 1087). Research ethical approval was granted by the Ethical Review Boards of the Centre de Recherche en Science Naturelle (CRSN) of Lwiro, DRC and the Kenya Medical Research Institute Scientific and Ethics Review Unit (KEMRI/SERU)-KEMRI/SERU/CGHR/005/3055, the Ethics committee of the Federal Capital Territory FHREC/2012/01/11/16-05-12 and the Ethics committee of Aminu Kano Teaching Hospital-NHREC/21/08/2008/AKTH/EC/1633 and the Health Services Management Board of Kano State- 2/1437AH-9/12/2015.

**Reporting summary**
Further information on research design is available in the Nature Portfolio Reporting Summary linked to this article.

## Data availability
Sequence data that support the findings of this study are available at SRA, accession IDs per isolate are available in Supplementary Data 1 and 3. All data generated during and/or analysed in the current study are available from the corresponding author on request. The exchange of biological material should always be in agreement with the local teams.

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

## Acknowledgements

We are grateful to Jacqueline Keane, Christoph Puethe and the Pathogen Informatics team (Wellcome Sanger Institute, Hinxton, Cambridge, United Kingdom) for the support. The work by S.V.P. and G.D. is funded in part by a grant from the Bill & Melinda Gates Foundation (OPP1151153). R.K. and M.B. were supported by research grants BB/N007964/1 and BB/ M025489/1, and by the BBSRC Institute Strategic Programme Microbes in the Food Chain BB/R012504/1 and its constituent projects BBS/E/F/ 000PR10348 and BBS/E/F/000PR10349. W.L.C. was supported by the Research Foundation—Flanders (FWO SB PhD fellowship 1S40018N); J.P.R. was financially supported by the Belgian Directorate General for Development Cooperation (DGD) M.A.B. and N.R.T. were supported by Wellcome funding to the Sanger Institute (#206194). The work done in Benin, Burkina Faso and DRC by B.B., L.M.-K., M.-F.P., D.F., D.A., J.J. and O.L. was funded by the Belgian Directorate of Development Cooperation (DGD) through the Multi-Year Programme (2012–2016) between the Belgian DGD and the Institute of Tropical Medicine, Belgium and (for DRC) by the Baillet-Latour find and the Flemish Interuniversity Council (VLIR-UOS). The isolates from Malawi were generated by Malawi Liverpool Wellcome Research Programme bacteraemia service, supported by Asia and Africa Programme Grant 206545/Z/17/Z to NF. The work in The Gambia was supported by the Bill & Melinda Gates Foundation (OPP1020327); GAVI The Vaccine Alliance's Accelerated Development and Introduction Plan (PneumoADIP), Medical Research Council (UK) to GM. *Salmonella* isolates obtained through the RTS,S study was funded by the Bill & Melinda Gates Foundation to CAM. *Salmonella* isolates obtained through the TSAP study were funded by the Bill & Melinda Gates Foundation to IVI (OPPGH5231) to F.M., H.J.J. and S.E.P. This research by S.V.P., S.S. and G.D. was funded by the National Institute for Health Research [Cambridge Biomedical Research Centre at the Cambridge University Hospitals NHS Foundation Trust]. The views expressed are those of the authors and not necessarily those of the NHS, the NIHR or the Department of Health and Social Care. This research was funded in whole, or in part, by the Wellcome Trust (#206194).

## Author contributions

Study design and oversight: S.V.P., G.D., Sa.K. and O.L. Isolate collection and metadata: B.B., H.J.J., L.M.-K., M.-F.P., D.F., D.M., O.V., D.A., J.P.R., P.-J.C., W.M., S.E.P., Si.K., K.O., J.P.A.L., J.R.M., S.A., A.S., S.O.A., K.P.A., W.O., L.T.O., M.C.T., P.L., I.F.H., T.M., C.M., F.-H.H., S.O., G.M., N.F., F.M., C.A.M., J.J., Sa.K. and O.L. Sequencing: S.V.P., Td.B., S.D. and N.R.T. Data analysis: S.V.P., Td.B., S.S., M.B., R.A.K., B.I., M.A.B., W.L.C., M.A.B.v.d.S. and N.R.T. Writing—original draft preparation: S.V.P. and G.D. Writing—a critical review: S.V.P., Td.B., S.S., M.B., R.A.K., B.I., M.A.B., B.B., H.J.J., L.M.-K., M.-F.P., D.F., D.M., O.V., D.A., J.P.R., P.-J.C., W.M., W.L.C., M.A.B.v.d.S., S.E.P., Si.K., K.O., J.P.A.L., J.R.M., S.A., A.S., S.O.A., K.P.A., W.O., L.T.O., M.C.T., P.L., I.F.H., T.M., C.M., F.H.-H., S.O., G.M., S.D., N.F., F.M., C.A.M., N.R.T., J.J., G.D., Sa.K. and O.L. All authors have read and agreed to the published version of the manuscript.

## Competing interests

The authors declare no competing interests.

## Additional information

Sandra Van Puyvelde [1,2,3] ✉, Tessa de Block[4], Sushmita Sridhar [1,2,5,6], Matt Bawn [7,8,9], Robert A. Kingsley [7,10], Brecht Ingelbeen [4,11], Mathew A. Beale [2], Barbara Barbé [4], Hyon Jin Jeon[1,12,13], Lisette Mbuyi-Kalonji [14,15], Marie-France Phoba[14,15], Dadi Falay[16], Delphine Martiny[17,18], Olivier Vandenberg [17,19], Dissou Affolabi[20], Jean Pierre Rutanga[4,21], Pieter-Jan Ceyssens[22], Wesley Mattheus[22], Wim L. Cuypers [4,23], Marianne A. B. van der Sande [4,11], Se Eun Park [12,24], Simon Kariuki[25], Kephas Otieno[25], John P. A. Lusingu[26,27], Joyce R. Mbwana [26], Samuel Adjei[28], Anima Sarfo[28], Seth O. Agyei[28], Kwaku P. Asante[29], Walter Otieno[30], Lucas Otieno[30], Marc C. Tahita [31], Palpouguini Lompo[31], Irving F. Hoffman[32], Tisungane Mvalo [32,33], Chisomo Msefula[34], Fatimah Hassan-Hanga[35,36], Stephen Obaro[37,38], Grant Mackenzie[39,40,41], Stijn Deborggraeve[4], Nicholas Feasey [32,42], Florian Marks [1,12,13,43], Calman A. MacLennan [44,45], Nicholas R. Thomson [2,40], Jan Jacobs [4,46], Gordon Dougan[1,48], Samuel Kariuki [47,48] & Octavie Lunguya[14,15,48]

[1]Cambridge Institute of Therapeutic Immunology and Infectious Disease, University of Cambridge School of Clinical Medicine, Cambridge Biomedical Campus, Cambridge CB2 0AW, UK. [2]Parasites and Microbes Programme, Wellcome Sanger Institute, Wellcome Genome Campus, Hinxton, Cambridge, UK. [3]Laboratory of Medical Microbiology, Vaccine & Infectious Disease Institute, University of Antwerp, Antwerp, Belgium. [4]Institute of Tropical Medicine, Antwerp, Belgium. [5]Division of Infectious Disease, Massachusetts General Hospital, Boston, MA, USA. [6]Department of Medicine, Harvard Medical School, Boston, MA, USA. [7]Quadram Institute Bioscience, Norwich, UK. [8]Earlham Institute, Norwich, UK. [9]Faculty of Biological Sciences, University of Leeds, Leeds, UK. [10]School of Biological Science, University of East Anglia, Norwich, UK. [11]Julius Center for Health Sciences and Primary Care, University Medical Center Utrecht, Utrecht University, Utrecht, the Netherlands. [12]International Vaccine Institute, 1 Gwanak-ro, Seoul 08826, Republic of Korea. [13]Madagascar Institute for Vaccine Research, University of Antananarivo, Antananarivo, Madagascar. [14]Department of Medical Biology, University Teaching Hospital of Kinshasa, Kinshasa, Democratic Republic of the Congo. [15]National Institute for Biomedical Research, Kinshasa, Democratic Republic of the Congo. [16]Department of Pediatrics, University Hospital of Kisangani, Kisangani, Democratic Republic of the Congo. [17]Department of Microbiology, Laboratoire Hospitalier Universitaire de Bruxelles-Universitair Laboratorium Brussel (LHUB-ULB), Université Libre de Bruxelles (ULB), 1000 Brussels, Belgium. [18]Faculty of Medicine and Pharmacy, University of Mons (UMONS), 7000 Mons, Belgium. [19]Division of Infection and Immunity, Faculty of Medical Sciences, University College London, London, UK. [20]Centre National Hospitalier Universitaire Hubert Koutoukou Maga, Cotonou, Benin. [21]College of Science and Technology, University of Rwanda, Kigali, Rwanda. [22]National Reference Center for Salmonella, Unit of Human Bacterial Diseases, Sciensano, J. Wytsmanstraat 14, B-1050 Brussels, Belgium. [23]Department of Computer Science, University of Antwerp, Antwerp, Belgium. [24]Yonsei University Graduate School of Public Health, Seodaemun-gu, Seoul 03722, Republic of Korea. [25]Kenya Medical Research Institute/Centre for Global Health Research, Kisumu, Kenya. [26]National Institute for Medical Research, Tanga, Tanzania. [27]Center for Medical Parasitology, Department of Immunology and Microbiology, University of Copenhagen, København, Denmark. [28]University of Health & Allied Sciences, Ho, Volta Region, Ghana. [29]Kintampo Health Research Centre, Research and Development Division, Ghana Health Service, Kintampo North Municipality, Ho, Volta Region, Ghana. [30]KEMRI/Walter Reed Project, Kombewa, Kenya. [31]Institut de Recherche en Science de la Santé, Direction Régionale du Centre-Ouest/ClinicalResearch Unit of Nanoro, Nanoro, Burkina Faso. [32]University of North Carolina Project, Lilongwe, Malawi. [33]Department of Pediatrics, School of Medicine, University of North Carolina at Chapel Hill, Chapel Hill, NC, USA. [34]Malawi Liverpool Wellcome Research Programme, Kamuzu University of Health Sciences, Blantyre, Malawi. [35]Department of Paediatrics, Bayero University, Kano, Nigeria. [36]Aminu Kano Teaching Hospital, Kano, Nigeria. [37]University of Nebraska Medical Center, Omaha, NE, USA. [38]International Foundation Against Infectious Diseases in Nigeria (IFAIN), Abuja, Nigeria. [39]Medical Research Council Unit The Gambia at London School of Hygiene & Tropical Medicine, Fajara, The Gambia. [40]London School of Hygiene and Tropical Medicine, Keppel St, Bloomsbury, London WC1E 7HT, UK. [41]Murdoch Children's Research Institute, Melbourne, VIC, Australia. [42]Department of Clinical Sciences, Liverpool School of Tropical Medicine, Liverpool, UK. [43]Heidelberg Institute of Global Health, University of Heidelberg, Heidelberg, Germany. [44]The Jenner Institute, Nuffield Department of Medicine, University of Oxford, Oxford, UK. [45]Enteric and Diarrheal Diseases, Global Health, Bill & Melinda Gates Foundation, Seattle, WA, USA. [46]Department of Microbiology, Immunology and Transplantation, KU Leuven, Leuven, Belgium. [47]Centre for Microbiology Research, Kenya Medical Research Institute, Nairobi, Kenya. [48]These authors contributed equally: Gordon Dougan, Samuel Kariuki, Octavie Lunguya. ✉e-mail: sandra.vanpuyvelde@uantwerpen.be

