## [Peer Review File · Nature Communications]

A genomic appraisal of invasive *Salmonella* Typhimurium and associated antibiotic resistance in sub-Saharan AfricaReviewers' comments:

Reviewer #1 (Remarks to the Author):

Previous studies have conducted similar genomic analyses at smaller scales and regional or national levels. In this paper, Van Puyvelde et al. aim to present a continental perspective of invasive *S. Typhimurium* and its antimicrobial resistance in sub-Saharan SA. They compared the prevalence, genotypes, and evolution of invasive *S. Typhimurium* obtained from 19 countries in sub-Saharan Africa. They also conducted temporal phylogenetic analyses to estimate the origins of specific invasive *S. Typhimurium* clades and investigated changes in genotypes and antimicrobial resistance over time.

While data reported in this paper are valuable due to limited information on phylogenomics of iNTS in Africa, some of the conclusions drawn by the authors may not be valid due to sampling bias. Specifically, the proportion of isolates from different countries varies substantially, which does not allow the analysis of regional changes in specific genotypes over time. Specifically, all 1980s isolates originate from Rwanda, whereas most isolates from 2014-2017 originate from Africa. Some of the conclusions that authors draw about dominant genotypes in specific periods are highly biased by the overrepresentation of isolates from particular countries; therefore, the findings could not be generalized to sub-Saharan Africa or East vs West Africa. While authors acknowledge the sampling bias as a limitation of this study in discussion (L533-542), they do not consider this when reporting some of the results (see specific comments below). The manuscript should be carefully revised to ensure that the conclusions are supported by the data.

Methods

L172: In addition to the maximum number of contigs, please report the median, min and max assembly N50, which reflects the fragmentation of a draft genome.

L178: It is unclear what "this plasmid sequences" means. Please also explain the criteria for selecting seven isolates for MinION sequencing.

L183 – 188 (and elsewhere): Please report parameters used for running individual programs or indicate when default parameters were used.

L289: Please report priors that were used for the substitution rate etc., and how they were

selected.

L326: Please clarify which isolates were tested using disk diffusion and how were disk diffusion data used, in addition to MICs.

L328: Clarify the goal of this statistical analysis (e.g., what were the groups that were compared?).

Results

L351: Clarify what kind of perspective you aimed to obtain.

L357: Please support this statement by reporting the number of genotypes identified in this study.

L368: Is it possible that the highest diversity of clades was identified in East African countries because sampling was more intensive in these countries? Would your statement still hold if you would randomly subsample the same number of isolates from each region?

L371: The 1980s samples predominantly originated from Rwanda and DRC. Since other countries were not sampled in the 1980s, it is not appropriate to state that iTYM2 and iTYM4 were dominant in 1980, as this misleads the reader to believe that they were dominant in sub-Saharan Africa as a whole and not only in Rwanda and DRC.

Fig. 1: It would be helpful if you could include another panel in Fig. 1 to show the proportion of isolates from each country each year, as this information is not immediately extractable from current figure panels.

L183: How reliable is the assignment of iTYM6's origin to DRC, given that the oldest (from 1980s) sSA isolates included in this study are from DRC and Rwanda but not from other sSA countries? Would your analysis results likely be different if older isolates from other sSA countries had been included? I understand that you may not have access to such isolates, but I believe this should be considered when interpreting the results.

L387: Were there any iTYM6 isolates from countries other than DRC and Rwanda included in the analyzed dataset? If not, could this have resulted in the lack of evidence from your analyses that iTYM6 was present anywhere outside the DRC until 1994?

Supplementary Table 1: Please list the clade and year of isolation for each isolate in the Supplementary table.

Fig. 3: It may be more appropriate to say "Estimated spread of iTYM6..." since there is missing data on iNTS isolates from multiple sSA countries, including countries neighbouring

DRC, which were estimated to be the sources of iTYM6.

L426: Could subtle but significant increases in ciprofloxacin MIC be due to increased expression of efflux pumps?

L427: What are these genes encoding, and how is their function related to ciprofloxacin resistance?

L430: It seems that “transmission” is missing in the title after “S. Typhimurium”.

L431: Consider revising the structure of this sentence.

Discussion

L529: What are the specific drivers of invasiveness discussed here?

Reviewer #2 (Remarks to the Author):

Over the past five years, Sandra Van Puyvelde and her team of researchers have led the field investigating the evolutionary complexity of the *Salmonella* Typhimurium variants responsible for causing iNTS bloodstream infections in Africa. Their comparative genomic approach has generated extremely important insights, largely focused on iNTS disease in the Democratic Republic of the Congo.

Here, the team have taken a broader approach to investigate *Salmonella* Typhimurium strains from 19 African countries, isolated over a 40 year period. This paper reports important discoveries relating to ESBL-mediated AMR and the emergence of XDR and pan-drug resistant (PDR) types of *Salmonella* Typhimurium, these are particular highlights of the current study.

The authors have built on the genotyping approach used to classify *Salmonella* Typhi that was developed by Vanessa Wong and Gordon Dougan in 2016 (PMID: 27703135). Such a genotyping scheme was needed because *Salmonella* Typhi variants could not be classified effectively by MLST, which is based on the allelic variation of 7 housekeeping genes. The whole-genome-based genotyping approach of Wong et al. 2016 has been widely accepted by the *Salmonella* Typhi community.

Here, the authors have used the same strategy as Wong et al. 2016 to develop a whole-genome-based genotyping approach for *Salmonella* Typhimurium isolates from Africa. The resulting phylogenetic analysis identified six major clades associated with African iNTS infections, named iTYM1 – iTYM6. This aspect of the study is problematic.

Previously, *Salmonella* Typhimurium isolates have been differentiated by the MLST approach. Sequence type 19 (ST19) has been shown to be responsible for the majority of cases of gastroenteritis worldwide. In contrast, the ST313 variant has been largely associated with bloodstream infection in Africa. The important take-home message from the last decades of research is that MLST classification is a robust way of differentiating *Salmonella* Typhimurium variants that cause human gastroenteritis from those that cause bloodstream infections.

The main problem with the suggestion that ST19 is split into clades iTYM2, iTYM3 and iTYM4 is that these new designations have not been put into context with previously published phylogenetic analyses of *S. Typhimurium* by Rob Kingsley's group (PMID: 32511244), Sam Miller's group (PMID: 26956590) and by Ruiting Lan's group (PMID: 28851865).

A similar problem is associated with the designation of ST313 into clades iTYM5 (ST313 Lineage I) and iTYM6 (ST313 Lineage II). Since the original discovery of ST313 in Africa in 2009 by Rob Kingsley and co-workers, 74 publications that involve the ST313 variant have appeared in PubMed. They largely focus on the replacement of ST313 Lineage I by ST313 Lineage II – and the new azithromycin-resistant ST313 sublineage II.1 reported by the Van Puyvelde team in 2019 (PMID: 31537784).

The proposed iTYM6 designation would “sever the connection” with the existing literature concerning ST313 Lineage II, and would cause confusion amongst the *Salmonella* research community.

I propose that the authors rethink their approach, and define the ST19 clades in the context of the existing literature - only adding new names when required. I also suggest that the

authors rethink their approach, and define the ST313 clades in the context of the existing literature - only adding new names when required.

Certain comments to be addressed.

It was not completely clear what the iTYM6 designation refers to. At lines 241 and 254, iTYM6 was defined as ST313 sublineage II.1. But at line 366, iTYM6 was defined as ST313 Lineage II.

At line 83 it is incorrectly stated that the authors have performed a “thorough scale appraisal of population structure of iNTS disease”. In fact the authors have only considered *S. Typhimurium*, and have ignored the 30% of iNTS cases that are caused by *Salmonella* Enteritidis.

At line 124, it is stated that the genomes of 1303 strains were used to provide a “comprehensive update”. However, as described in the methods, the genomes of 816 of these isolates have been published previously. The section describing the key findings (starting at line 364) implies that the samples were systematically collected in some way. In fact, the authors have done a great deal of “freezer surfing” to obtain isolates - so the analysis reflects strains that were stored by various researchers, and do not accurately reflect incidence in different parts of Africa. This limitation needs to be stated clearly.

At line 235, the reasoning for adapting the *S. Typhi* classification system to *S. Typhimurium* is presented. However, as discussed above, *S. Typhimurium* presents a different case as MLST has already proved to be a useful discriminator.

At line 364 it is stated that clades associated with invasive *S. Typhimurium* infections have been identified. This statement ignores the fact that 7.6 percent of the *S. Typhimurium* isolates were derived from stool.

At line 371, as described above, the discussion of clades iTYM2 & iTYM4 lacks context because previous literature has not been referred to.

Line 391, it is stated that there were five independent introductions of *S. Typhimurium* into

the East African region. The scientific basis of these deductions is not explained.

The important section that describes the evolutionary processes (starting at line 477) ignores much important literature that has described pseudogenes in African *S. Typhimurium* over the past few years. How many of the pseudogenes described in this study have been identified before? These should be named clearly.

Figure 1 panel C appears to show the number of isolates that were used in this study. However, this is not stated in the legend.

The important figure 2 shows the relationship between the different clades of *S. Typhimurium* ST19 and ST313. However, this is not connected with the existing literature. Contextual isolates such as A130, D23580 and the author's own sublineage II.1 should be highlighted on the tree.

Figure 3A present important phylogeographic however, these are meaningless without labelling the different clades. Again, contextual isolates such as A130, D23580 and the author's own sublineage II.1 should be highlighted on the tree. The reasoning used to deduce the spread of iTYM6 across Africa shown in Figure 3B needs to be carefully articulated and justified.

Figures 5 & 7 would be much improved by using a more "readable" font such as Arial.

Are any of the plasmids shown in Figure 6 related to the pST313 plasmid that was reported in 2015 (PMID: 25779570), and shares the same incompatibility group?

Figure 9B is very difficult to understand. What do clades alpha and beta refer to? How can the reader determine exactly which isolates were used for this analysis?

The genes listed in Table 3 need to be put into context with the previous literature.

The units used for the scale bar for the phylogenetic trees in the manuscript must be

defined.

In Supplementary figure 4, it is not clear which of the exterior rings is which. The rings that represent country or clusters need to be labelled clearly. Several of the colours used for the different countries are too similar in colour, and do not allow countries to be effectively distinguished.

Well-characterised plasmids should be added to the tree shown in Supplemental Figure 9 to allow interpretation in the context of the literature. For example, where is the the pST313 plasmid that was reported in 2015 (PMID: 25779570)?

Reviewer #3 (Remarks to the Author):

The manuscript, "A genomic appraisal of invasive Salmonella Typhimurium and associated antibiotic resistance in sub-Saharan Africa", by Van Puyvelde et al, is an in depth genomic characterization of invasive Salmonella Typhimurium. The authors put their new data in the context of previous studies in the region and show improvement of models identifying emergence of particular strains of S. Typhimurium. The authors also characterize antibiotic resistance associated with these isolates and apply long-read technology to map out critical plasmids that confer AR in the region. Some limitations to the study, that the authors also identify, is that surveillance for iNTS is limited in Africa so the sample most likely is over-represented by isolates related to outbreaks or special studies. This may affect some of the estimates of strain emergence and AR.

Specific feedback:

1. Lines 168-175 (genome sequencing section) – Did the authors apply any quality filters to the sequence data used in this study including coverage, read quality score, etc. The authors mention a quality threshold for assembly and contamination but not for the reads themselves.
2. Lines 215-216 – the authors mention minimum thresholds then use the lesser than (<) symbol in the text, do they not mean greater than (>)
3. Lines 228, do the authors filter out invariant sites before constructing their phylogeny? Why do the authors use different methods, different references, and different SNP callers

between the phylogenetic analysis section and the evolutionary context analysis?

4. Did the authors try different clustering methods like single linkage or minimum spanning trees for their identification of clades and subclades for the iTYM clades? In line 245 the authors mention hierarchical clustering is a phylogeny free approach but then previously mention clades and subclades used for classification which implies a phylogeny was generated. Can the authors clarify what they mean?

5. Line 352 - It's not clear to me the importance of the non-African isolates since most of them do not sit within any of the lineages identified. Please highlight in the text how many non-African genomes fell within the 6 African lineages.

6. Results section 1 - The isolates listed for this study were from diverse sources including stool, urine, and an animal. Were all isolates, regardless of source, considered invasive? If so, how were stool isolates identified as invasive versus gastrointestinal?

7. Line 362/supplemental figure 2. There is an MLST label over the source type info on the iTol tree. Also how are the 71k SNPs identified, is this only variable sites, are non-variable or invariant sites filtered out? Also where is the scale bar (I don't see a scale bar on any of the supplemental iTol trees)?

8. Sup figure 3 and paper - there are more heirbaps clusters than named. Can the authors mention why they decided to focus on the particular clusters they identified as the 6 key ones.

9. Line 375 - It looks like the 6 clade isolates emerged as the predominate clade in 2001 and continue to represent the most dominate clade, not peaked in 2001 as is mentioned in the text

10. Line 381. What were the criteria for representative isolates? A phylogenetic tree is shown as part of the selection process but since this is for a BEAST analysis, was year of collection also taken into consideration? Suggest considering adding year as one of the rings in the iTol tree to represent that variable.

11. Figure 3A Suggest putting the confidence intervals on Figure 3a to denote the likely range for when the strain emerged. For figure 3A - please include the colors in the legend, I'm assuming pink is the UK but it's not listed anywhere?

12. Line 424, Since MICS are measured at discreet values a median would be more informative than mean. Additionally, these results are puzzling that resistance would change with further nested clades on a phylogenetic tree. If this is generated from the subset of

data, can the larger data set be looked at to see if other isolates that fall in these clades have a similar phenomenon?

13. For Figure 5, can the plasmid types be listed in the figure – IncHI1 IncHI2 or IncI

14. Not sure the value of Figure 8 of how the small peaks of iTYM6.3 and 6.7 represent an outbreak, since these represent small peaks compared to all iTYM clades? Were these isolates collected specifically during ongoing outbreaks? Maybe show these subclades against a background of isolates just from DRC? Also can consider moving this to the supplemental.

15. Line 522 – Another limitation of the study is the bias the data set has for the start of the time period and the end of the time period of this study. This may impact some of the estimates for strain emergence.

We would like to thank the reviewers for their in-depth review of our manuscript.

Reviewers' comments:

Reviewer #1 (Remarks to the Author):

Methods

L172: In addition to the maximum number of contigs, please report the median, min and max assembly N50, which reflects the fragmentation of a draft genome.

Thanks for this comment. Detailed quality information on the assemblies and reads have been included to this manuscript as a Supplementary Dataset 1.

L178: It is unclear what “this plasmid sequences” means. Please also explain the criteria for selecting seven isolates for MinION sequencing.

This has been included in the main text at L205.

L183 – 188 (and elsewhere): Please report parameters used for running individual programs or indicate when default parameters were used.

This has been included in the main text.

L289: Please report priors that were used for the substitution rate etc., and how they were selected.

We performed a spatiotemporal analysis in line with earlier studies of invasive *S. Typhimurium* from sub-Saharan Africa (Okoro et al., 2012, Nature Genetics and Van Puyvelde et al., 2019, Nature Communications). We compared different models (including evaluating a strict and relaxed clock rate, a constant, exponential, and Bayesian Skyline model of population size) and used pathfinding and stepping stone analysis to evaluate these with a Bayes factor. As described in the Methods, a general time reversible (GTR) substitution model with diffuse gamma distribution prior (shape 0.001, scale 1,000) and invariant sites, an uncorrelated log-normal relaxed molecular clock and a model with exponential population size was identified to be optimal over 80 million Markov Chain Monte Carlo (MCMC) cycles.

L326: Please clarify which isolates were tested using disk diffusion and how were disk diffusion data used, in addition to MICs.

The disk diffusion tests were only performed in the lab for confirmation. As we agree with the reviewer that this causes confusion and as only the E-test results were used in this manuscript, this was updated in the Methods section.

L328: Clarify the goal of this statistical analysis (e.g., what were the groups that were compared?).

This was clarified in the Methods section. The definition of the different groups are further specified in Supplementary Figure 2A and a reference to that figure is made in the Methods section.

Results

L351: Clarify what kind of perspective you aimed to obtain.

This has been added at the sentence at L425.

L357: Please support this statement by reporting the number of genotypes identified in this study.

Thanks for the question, this has been included at L432.

L368: Is it possible that the highest diversity of clades was identified in East African countries because sampling was more intensive in these countries? Would your statement still hold if you would randomly subsample the same number of isolates from each region?

Thank you for this question.

The sample collection is indeed opportunistic, given that data from iNTS is sparse.

The effect of sample bias should be throughout the manuscript. We have included all available data to reach the most optimal view possible and used rigorous statistical methods to analyse these. However, subsampling would not solve the possible problem of missing data but decrease the resolution of the results obtained with the data we have now. The scarcity of isolates has been discussed at L869, and this specific concern is added to that section.

L371: The 1980s samples predominantly originated from Rwanda and DRC. Since other countries were not sampled in the 1980s, it is not appropriate to state that iTYM2 and iTYM4 were dominant in 1980, as this misleads the reader to believe that they were dominant in sub-Saharan Africa as a whole and not only in Rwanda and DRC.

This has been adapted in the text

Fig. 1: It would be helpful if you could include another panel in Fig. 1 to show the proportion of isolates from each country each year, as this information is not immediately extractable from current figure panels.

Thanks for the suggestion and we agree with the reviewer that this is valuable information of this manuscript. The suggested data is already available in Supplementary Figure 1 but a reference to that figure was missing at the respective location. Therefore, the reference to the Supplementary Figure 1 has been included in the main text, at the same place as a reference to Figure 1.

L183: How reliable is the assignment of iTYM6's origin to DRC, given that the oldest (from 1980s) sSA isolates included in this study are from DRC and Rwanda but not from other sSA countries? Would your analysis results likely be different if older isolates from other sSA countries had been included? I understand that you may not have access to such isolates, but I believe this should be considered when interpreting the results.

Thank you for this interesting question. This is indeed a limitation of our data, and inaccurate source attribution in the face of sampling bias is a challenge that can be difficult to overcome. We only had access to historical isolates from DRC and Rwanda, and we are only aware of historical isolates originating from these regions. It is through the pioneering work of late J. Vandepitte and his colleagues in these regions that we know of invasive *Salmonella* infections from the 1980s onwards in DRC and have access to these data. Some of these early reports were in French, which we explored in addition to the Pubmed repository. Regretably, we did not have access to any additional samples that might further inform this important question.

With all given data, the DRC as the origin is however the best prediction we can make. The Bayesian algorithms (BEAST) take all available information into account to come to this prediction. Note that additionally not only the MRCA of iTYM6/ST313-L2 as a whole is predicted to originate from the DRC, but also the ancestry nodes underlying the different substructures in the tree, adding confidence to our observation.

Although the literature is sparse, invasive *Salmonella* infections by ST313-L2 have not been found from elsewhere and we are not aware of other isolates that could have been included in the analysis.

We have now added a section in our Discussion where we describe this potential limitation in our work L883.

L387: Were there any iTYM6 isolates from countries other than DRC and Rwanda included in the analyzed dataset? If not, could this have resulted in the lack of evidence from your analyses that iTYM6 was present anywhere outside the DRC until 1994?

The selection for this dataset is described in our material and methods at L342 and aimed to include optimal coverage of spatiotemporal variation in the dataset, whilst avoiding over enrichment for samples from the DRC and Rwanda. The selected dataset is also visualised on Supplementary Figure 6, and includes genomes from 18 African countries.

Supplementary Table 1: Please list the clade and year of isolation for each isolate in the Supplementary table.

Thank you for the suggestion, the clades and subclades have been added to Supplementary Table 1, whilst the year of isolation was already part of that table.

Fig. 3: It may be more appropriate to say “Estimated spread of iTYM6...” since there is missing data on iNTS isolates from multiple sSA countries, including countries neighbouring DRC, which were estimated to be the sources of iTYM6.

Thank you, we have amended this.

L426: Could subtle but significant increases in ciprofloxacin MIC be due to increased expression of efflux pumps?

Thanks for the suggestions and this is indeed possible. We have included this hypothesis at L768.

L427: What are these genes encoding, and how is their function related to ciprofloxacin resistance?

This has been included a section in the main text on the function of these genes at L765.

L430: It seems that “transmission” is missing in the title after “S. Typhimurium”.

Thanks for the suggestions. The suggested change could however cause confusion as transmission is often used in other contexts for iNTS. Therefore, alternatively, we decided to rephrase the title to: XDR and PDR invasive *S. Typhimurium* is associated with the presence of IncHI2 and IncI1 plasmids

L431: Consider revising the structure of this sentence.

This sentence has been revised.

Discussion

L529: What are the specific drivers of invasiveness discussed here?

These are indeed unknown and therefore, the sentence has been updated to reflect this better.

Reviewer #2 (Remarks to the Author):

Here, the authors have used the same strategy as Wong et al. 2016 to develop a whole-genome-based genotyping approach for *Salmonella Typhimurium* isolates from Africa. The resulting phylogenetic analysis identified six major clades associated with African iNTS infections, named iTYM1 – iTYM6. This aspect of the study is problematic.

Previously, *Salmonella* Typhimurium isolates have been differentiated by the MLST approach. Sequence type 19 (ST19) has been shown to be responsible for the majority of cases of gastroenteritis worldwide. In contrast, the ST313 variant has been largely associated with bloodstream infection in Africa. The important take-home message from the last decades of research is that MLST classification is a robust way of differentiating *Salmonella* Typhimurium variants that cause human gastroenteritis from those that cause bloodstream infections.

The main problem with the suggestion that ST19 is split into clades iTYM2, iTYM2, iTYM3 and iTYM4 is that these new designations have not been put into context with previously published phylogenetic analyses of *S. Typhimurium* by Rob Kingsley's group (PMID: 32511244), Sam Miller's group (PMID: 26956590) and by Ruiting Lan's group (PMID: 28851865).

A similar problem is associated with the designation of ST313 into clades iTYM5 (ST313 Lineage I) and iTYM6 (ST313 Lineage II). Since the original discovery of ST313 in Africa in 2009 by Rob Kingsley and co-workers, 74 publications that involve the ST313 variant have appeared in PubMed. They largely focus on the replacement of ST313 Lineage I by ST313 Lineage II – and the new azithromycin-resistant ST313 sublineage II.1 reported by the Van Puyvelde team in 2019 (PMID: 31537784).

The proposed iTYM6 designation would “sever the connection” with the existing literature concerning ST313 Lineage II, and would cause confusion amongst the *Salmonella* research community.

I propose that the authors rethink their approach, and define the ST19 clades in the context of the existing literature - only adding new names when required. I also suggest that the authors rethink their approach, and define the ST313 clades in the context of the existing literature - only adding new names when required.

This is an interesting and important point to address, and we thank the reviewer for their constructive input.

We believe that using the MLST alone to classify between invasive and non-invasive infections would however be problematic. First, ST313 isolates are indeed very often linked to invasiveness, while ST19 isolates are often linked to non-invasiveness but this association is not complete and in line with the recent findings. It is one of the main findings of our paper, supported by recent literature, that invasive lineages exist in ST19 as well (Kariuki et al. PMID 32745137). Second, the distinction between ST19 and ST313 is not a robust classifier for invasiveness. ST313 is one SNP different from ST19, which is a SNP with no biological meaning for invasiveness. Taking the full population structure into account is better at acknowledging the complexity of evolution towards invasiveness and is therefore preferable.

The raised comment however reflects the urgent need for a sustainable and clear nomenclature of invasive *S. Typhimurium*, taking the complex classification between invasiveness and non-invasiveness into account, which we aim to address here.

We agree with reviewer 2 that our suggested nomenclature can be better linked to the MLST-based *Salmonella* nomenclature to allow linkage to the existing literature.

Therefore we propose to use a hybrid naming based on the current naming of ST313 lineages. We changed the names to ST313-L1, ST313-L2 (instead of iTYM5-6), and extend with ST19-L1 to ST19-L4 (instead of iTYM1-4) throughout the manuscript. This is better in line with the existing literature on Pubmed and sustainable towards the future when new lineages would become identified. We also preferred to use Arabic instead of Roman numbers, as both are used in the literature and as higher Roman numbers are more difficult to interpret and cause confusion.

Certain comments to be addressed.

It was not completely clear what the iTYM6 designation refers to. At lines 241 and 254, iTYM6 was defined as ST313 sublineage II.1. But at line 366, iTYM6 was defined as ST313 Lineage II.

Thanks for the suggestion, this has been clarified. iTYM6 is changed to ST313-lin2 throughout the manuscript.

At line 83 it is incorrectly stated that the authors have performed a “thorough scale appraisal of population structure of iNTS disease”. In fact the authors have only considered *S. Typhimurium*, and have ignored the 30% of iNTS cases that are caused by *Salmonella* Enteritidis.

The sentence was adapted to take this into account.

At line 124, it is stated that the genomes of 1303 strains were used to provide a “comprehensive update”. However, as described in the methods, the genomes of 816 of these isolates have been published previously. The section describing the key findings (starting at line 364) implies that the samples were systematically collected in some way. In fact, the authors have done a great deal of “freezer surfing” to obtain isolates - so the analysis reflects strains that were stored by various researchers, and do not accurately reflect incidence in different parts of Africa. This limitation needs to be stated clearly.

This specific sentence was adapted to avoid confusion on this. The limitation on sample scarcity has been discussed above, and was further elaborated in the discussion (from L866). We have used the most optimal dataset available in combination with robust statistical methods to maximally mitigate these effects.

At line 235, the reasoning for adapting the *S. Typhi* classification system to *S. Typhimurium* is presented. However, as discussed above, *S. Typhimurium* presents a different case as MLST has already proved to be a useful discriminator.

Thanks for the comment. As discussed above, the MLST classification of *S. Typhimurium* is important but not the best discriminator between invasiveness and non-invasiveness. This is now better explained in the introduction of the manuscript at L119 for clarity.

At line 364 it is stated that clades associated with invasive *S. Typhimurium* infections have been identified. This statement ignores the fact that 7.6 percent of the *S. Typhimurium* isolates were derived from stool.

We know from earlier work that within the African setting, there is an epidemiological link between carriage in the gut and invasive disease in endemic regions. Two major studies have shown carriage of invasive clones in the gut within such endemic settings. Kariuki et al (2020) showed circulation of the same isolates in a slum in Nairobi among healthy children and children with bloodstream infections. This shows that within the specific setting, the invasive isolates are carried by the larger population. Post et al. (2019) showed that household members of a child with invasive disease carried the same isolate as the patient. 61 of the 84 African faeces isolates originate from these two settings and can thus be considered to be representatives of invasive disease. This reasoning has been added to the manuscript at L185.

At line 371, as described above, the discussion of clades iTYM2 & iTYM4 lacks context because previous literature has not been referred to.

Thank you for the suggestion. These clades have been linked to the literature in the discussion section. iTYM2 (ST313-L2) has not been observed before, while iTYM4 has been observed recently in Kenya. A reference to this study is included at L841.

Line 391, it is stated that there were five independent introductions of *S. Typhimurium* into the East African region. The scientific basis of these deductions is not explained.

These are the observations based on the BEAST analysis. The interpretation to come to this conclusion is further explained in the methodology at L366.

The important section that describes the evolutionary processes (starting at line 477) ignores much important literature that has described pseudogenes in African *S. Typhimurium* over the past few years. How many of the pseudogenes described in this study have been identified before? These should be named clearly.

Thank you for the suggestion. These pseudogenes have been searched in the literature, and a section on this has been included at L765.

Figure 1 panel C appears to show the number of isolates that were used in this study. However, this is not stated in the legend.

Thank you for this comment, this has been added.

The important figure 2 shows the relationship between the different clades of *S. Typhimurium* ST19 and ST313. However, this is not connected with the existing literature. Contextual isolates such as A130, D23580 and the author's own sublineage II.1 should be highlighted on the tree.

Thank you for the suggestion. The contextual isolates D23580 and 10433_3 have been highlighted on the tree and the legend refers additionally to ST313-L2 subclade 3 which coincides with the previously called sublineage II.1. Note that isolate A130 of ST313-L1 is not part of the ST313-L2 branch of the *S. Typhimurium* population analysed in figure 2.

Figure 3A present important phylogeographic however, these are meaningless without labelling the different clades. Again, contextual isolates such as A130, D23580 and the author's own sublineage II.1 should be highlighted on the tree. The reasoning used to deduce the spread of iTYM6 across Africa shown in Figure 3B needs to be carefully articulated and justified.

Isolate D23580 is annotated on figure 3A. Additionally, all the ST313-L2 subclades (as presented in figure 2) are annotated too. Sublineage II.1 coincides with subclade 3. As discussed above, the interpretation and reasoning behind figure 3B is explained in material and methods.

Figures 5 & 7 would be much improved by using a more "readable" font such as Arial.

The font of figures 5 and 7 is set to Arial.

Are any of the plasmids shown in Figure 6 related to the pST313 plasmid that was reported in 2015 (PMID: 25779570), and shares the same incompatibility group?

Yes, the respective references were added to Figure 6.

Figure 9B is very difficult to understand. What do clades alpha and beta refer to? How can the reader determine exactly which isolates were used for this analysis?

Thank you for this comment. The Figure legend, and corresponding section in the results section starting at L771 have been rewritten to make this more clear. As explained in material and methods, the additional isolates used for this analysis are listed in Supplementary Table 4. A new Supplementary Table 3 is included, listing dataset of 10 isolates per invasive *S. Typhimurium* clade so that it is clear which isolates were included exactly in this analysis.

The genes listed in Table 3 need to be put into context with the previous literature.

Thank you for the suggestion. All genes from Table 3 have been put in context with previous literature, and this was added to the table. We suggest however moving table 3 to supplementary material (Supplementary Table 5), to keep the main manuscript focused and to keep a reasonable number of references in the main text.

The units used for the scale bar for the phylogenetic trees in the manuscript must be defined.

This is included on all trees.

In Supplementary figure 4, it is not clear which of the exterior rings is which. The rings that represent country or clusters need to be labelled clearly. Several of the colours used for the different countries are too similar in colour, and do not allow countries to be effectively distinguished.

This has been added in the legend of the figure, in compliance with all other figures of the manuscript.

Well-characterised plasmids should be added to the tree shown in Supplemental Figure 9 to allow interpretation in the context of the literature. For example, where is the the pST313 plasmid that was reported in 2015 (PMID: 25779570)?

Thank you for the suggestion, these have been included.

Reviewer #3 (Remarks to the Author):

Specific feedback:

1. Lines 168-175 (genome sequencing section) – Did the authors apply any quality filters to the sequence data used in this study including coverage, read quality score, etc. The authors mention a quality threshold for assembly and contamination but not for the reads themselves.

Thanks for this question and this was done. The quality of reads was checked using the internal Sanger sequencing pipelines. This includes a variety of parameters such as check of the presence of adapters, contamination level, GC fraction, insert size, qX yield, match with reference genomes, coverage of mapping. All isolates passed quality check. This has been added in the main text at L196.

2. Lines 215-216 – the authors mention minimum thresholds then use the lesser than (<) symbol in the text, do they not mean greater than (>)

Thanks for this comment. We have adapted this.

3. Lines 228, do the authors filter out invariant sites before constructing their phylogeny? Why do the authors use different methods, different references, and different SNP callers between the phylogenetic analysis section and the evolutionary context analysis?

Yes, this is being done with snp-sites (L235). A slightly different methodology has been used for the phylogenetic analysis and the evolutionary context analysis, in line with the previous literature and as the datasets have different characteristics in terms of diversity. The phylogenetic analysis is optimized to reconstruct a high-resolution phylogeny, whereas the evolutionary context analysis aims at placing the isolates in a more diverse set of strains.

4. Did the authors try different clustering methods like single linkage or minimum spanning trees for their identification of clades and subclades for the iTYM clades? In line 245 the authors mention hierarchal clustering is a phylogeny free approach but then previously mention clades and subclades

used for classification which implies a phylogeny was generated. Can the authors clarify what they mean?

We have chosen to use hierBAPS based on the earlier *Salmonella* literature (Wong et al., Nature Communications, 2016). HierBAPS itself is indeed a phylogeny-free approach, and the clades and subclades were identified here by mapping the hierBAPS results on our RAxML tree. This has been adjusted at L285 to clarify.

5. Line 352 - It's not clear to me the importance of the non-African isolates since most of them do not sit within any of the lineages identified. Please highlight in the text how many non-African genomes fell within the 6 African lineages.

Thank you for the suggestion. To make these data available for the reader, an additional column has been added to Supplementary Table 1, including the clades and subclades – together with metadata such as origin and year of isolation. The number of non-African isolates in the clades has been added in the main text at L295.

6. Results section 1 - The isolates listed for this study were from diverse sources including stool, urine, and an animal. Were all isolates, regardless of source, considered invasive? If so, how were stool isolates identified as invasive versus gastrointestinal?

Thanks for this comment, we have addressed this comment as discussed above.

7. Line 362/supplemental figure 2. There is an MLST label over the source type info on the iTol tree. Also how are the 71k SNPs identified, is this only variable sites, are non-variable or invariant sites filtered out? Also where is the scale bar (I don't see a scale bar on any of the supplemental iTol trees)?

We assume that the reviewer means supplementary figure 3.

Thank you for the suggestion. We have adapted the MLST label to source. The tree is identified on the SNPs as elaborated in material and methods. The scale bar is at the top of the figure (called 'tree scale').

8. Sup figure 3 and paper - there are more heirbaps clusters than named. Can the authors mention why they decided to focus on the particular clusters they identified as the 6 key ones.

We assume that the reviewer means supplementary figure 4.

This is explained in the material and methods under the section "Identification of clades and subclades". We believe that there is confusion between 'clusters' coinciding with HierBaps groups, with the 'clades' which were identified in this study.

9. Line 375 - It looks like the 6 clade isolates emerged as the predominate clade in 2001 and continue to represent the most dominate clade, not peaked in 2001 as is mentioned in the text

Thank you for the suggestion. This sentence has been rewritten.

10. Line 381. What were the criteria for representative isolates? A phylogenetic tree is shown as part of the selection process but since this is for a BEAST analysis, was year of collection also taken into consideration? Suggest considering adding year as one of the rings in the iTol tree to represent that variable.

As described in our Methods, the dataset was indeed first stratified by year. Since the tree presented in Figure 3A is timescaled, the year of sampling is already shown by the position of the tips along the x axis.

11. Figure 3A Suggest putting the confidence intervals on Figure 3a to denote the likely range for when the strain emerged. For figure 3A - please include the colors in the legend, I'm assuming pink is the UK but it's not listed anywhere?

Thank you for this suggestion. Whilst we agree with the reviewer that showing the confidence intervals on the figure in the main manuscript would be preferable, doing so renders this figure too busy to allow interpretation. Therefore, a separate figure with confidence intervals is included as Supplementary material. A color legend has been included on figure 3A.

12. Line 424, Since MICS are measured at discreet values a median would be more informative than mean. Additionally, these results are puzzling that resistance would change with further nested clades on a phylogenetic tree. If this is generated from the subset of data, can the larger data set be looked at to see if other isolates that fall in these clades have a similar phenomenon?

Thanks for the comment. The median values for all subclades were given as part of the boxplots of Supplementary Figure 2 and these values have been included in the main text at L603. There was no pruning of isolates among this subtree, and all nested isolates are already part of the analysis.

13. For Figure 5, can the plasmid types be listed in the figure – IncHI1 IncHI2 or IncI

The type of plasmid has been added in the figure. The full plasmid name and all associated information is available as the information from table 2. Therefore a reference to this table has been made in the legend of figure 5.

14. Not sure the value of Figure 8 of how the small peaks of iTYM6.3 and 6.7 represent an outbreak, since these represent small peaks compared to all iTYM clades? Were these isolates collected specifically during ongoing outbreaks? Maybe show these subclades against a background of isolates just from DRC? Also can consider moving this to the supplemental.

We would like to note that the graph indeed shows the number of isolates from the Kisantu region in DRC, as noted in the title of the figure. The isolates were collected during ongoing surveillance, and are among the most resistant iNTS strains identified yet. This figure has been moved to supplementary material as suggested.

15. Line 522 – Another limitation of the study is the bias the data set has for the start of the time period and the end of the time period of this study. This may impact some of the estimates for strain emergence.

We thank the reviewer for this comment and refer to the earlier discussion on this subject above.

REVIEWERS' COMMENTS

Reviewer #1 (Remarks to the Author):

The authors have adequately addressed comments.

Reviewer #2 (Remarks to the Author):

It was a pleasure to read the revised version of this important manuscript. The authors have addressed my concern that the proposed iTYM6 designation would “sever the connection” with the existing literature concerning ST313 Lineage II, and would cause confusion amongst the Salmonella research community. It is good to see that the ST19 and ST3131 clades have now been considered in the context of the existing literature, and named appropriately.

A few minor comments to be addressed.

Line 413: add a few words to explain the evidence for the “five independent introductions”.

Lines 522 – 528: this paragraph should be rephrased to make it easier to understand.

On page 39, please change the font colour for the blue rectangles labelled “Malawi 2004”, “Malawi 1996”, and “Mozambique 1999” - from a black font to a white font.

On page 47, please change the “lineage I” and “lineage II” labels to be consistent with figure 3. Change Lineage I to Lineage 1, and Lineage II to lineage 2.

Reviewer #3 (Remarks to the Author):

The authors adequately addressed previous comments provided for the initial review of the manuscript. No further comments at this time.

We would like to thank the reviewers for their final comments on our manuscript.

REVIEWERS' COMMENTS

Reviewer #1 (Remarks to the Author):

The authors have adequately addressed comments.

Reviewer #2 (Remarks to the Author):

It was a pleasure to read the revised version of this important manuscript. The authors have addressed my concern that the proposed iTYM6 designation would “sever the connection” with the existing literature concerning ST313 Lineage II, and would cause confusion amongst the Salmonella research community. It is good to see that the ST19 and ST3131 clades have now been considered in the context of the existing literature, and named appropriately.

A few minor comments to be addressed.

Line 413: add a few words to explain the evidence for the “five independent introductions”.

This has been included.

Lines 522 – 528: this paragraph should be rephrased to make it easier to understand.

The paragraph has been rewritten.

On page 39, please change the font colour for the blue rectangles labelled “Malawi 2004”, “Malawi 1996”, and “Mozambique 1999” - from a black font to a white font.

This has been edited.

On page 47, please change the “lineage I” and “lineage II” labels to be consistent with figure 3. Change Lineage I to Lineage 1, and Lineage II to lineage 2.

This has been edited.

Reviewer #3 (Remarks to the Author):

The authors adequately addressed previous comments provided for the initial review of the manuscript. No further comments at this time.